# Major patterns in the introgression history of *Heliconius* butterflies

**Yuttapong Thawornwattana[1]\*, Fernando Seixas[1], Ziheng Yang[2]\*, James Mallet[1]\***

[1]Department of Organismic and Evolutionary Biology, Harvard University, Cambridge, United States; [2]Department of Genetics, Evolution and Environment, University College London, London, United Kingdom

**\*For correspondence:**
ythaworwattana@g.harvard.edu (YT);
z.yang@ucl.ac.uk (ZY);
jmallet@oeb.harvard.edu (JM)

**Competing interest:** The authors declare that no competing interests exist.

**Abstract** Gene flow between species, although usually deleterious, is an important evolutionary process that can facilitate adaptation and lead to species diversification. It also makes estimation of species relationships difficult. Here, we use the full-likelihood multispecies coalescent (MSC) approach to estimate species phylogeny and major introgression events in *Heliconius* butterflies from whole-genome sequence data. We obtain a robust estimate of species branching order among major clades in the genus, including the 'melpomene-silvaniform' group, which shows extensive historical and ongoing gene flow. We obtain chromosome-level estimates of key parameters in the species phylogeny, including species divergence times, present-day and ancestral population sizes, as well as the direction, timing, and intensity of gene flow. Our analysis leads to a phylogeny with introgression events that differ from those obtained in previous studies. We find that *Heliconius aoede* most likely represents the earliest-branching lineage of the genus and that 'silvaniform' species are paraphyletic within the melpomene-silvaniform group. Our phylogeny provides new, parsimonious histories for the origins of key traits in *Heliconius*, including pollen feeding and an inversion involved in wing pattern mimicry. Our results demonstrate the power and feasibility of the full-likelihood MSC approach for estimating species phylogeny and key population parameters despite extensive gene flow. The methods used here should be useful for analysis of other difficult species groups with high rates of introgression.

## eLife assessment

This **important** study revises the evolutionary history of *Heliconius* butterflies, a well-established model system for understanding speciation in the presence of gene flow between species. Using a **convincing** statistical phylogenetic approach that relies on the multispecies coalescent, the authors reconstruct the evolution of the lineage, including the timing of speciation events and the history of gene flow. The new phylogeny will be of interest to all researchers working on *Heliconius* butterflies, and the phylogenetic approach to investigators aiming to understand the history of lineages that have experienced extensive gene flow.

## Introduction

Introgression among species has been reported in a wide range of organisms, including humans and Neanderthals (*Kuhlwilm et al., 2016*), Darwin's finches (*Lamichhaney et al., 2018*), felids (*Li et al., 2016*; *Li et al., 2019*), canids (*Gopalakrishnan et al., 2018*), horses (*Jónsson et al., 2014*; *Gaunitz et al., 2018*), living and extinct elephants (*Palkopoulou et al., 2018*), cichlid fishes (*Malinsky et al., 2018*), malaria mosquitoes (*Fontaine et al., 2015*; *Small et al., 2020*), *Drosophila* fruit flies (*Suvorov et al., 2022*), sunflowers (*Rieseberg et al., 2003*; *Whitney et al., 2010*), maize (*Calfee et al., 2021*), and yeast (*Leducq et al., 2016*). Introgression is now widely recognized as an important process that

can facilitate adaptation and speciation. While previous studies have confirmed the prevalence of gene flow, they give only a fragmented understanding of introgression because they mostly rely on approximate methods based on simple data summaries such as genome-wide site pattern counts or estimated gene trees. Those methods have limited power and cannot infer certain modes of gene flow or estimate most population parameters characterizing the process of species divergence and gene flow.

Recent advances make it possible to model genomic evolution under the multispecies coalescent (MSC) framework and use full-likelihood methods to estimate the species tree and quantify introgression (*Wen and Nakhleh, 2018*; *Zhang et al., 2018*; *Flouri et al., 2020*). Analyses of both simulated and real data demonstrate that this full-likelihood MSC approach is efficient, accurate, and robust to moderate levels of model violation (*Huang et al., 2022*; *Thawornwattana et al., 2023*). A major advantage over approximate methods is the ability to estimate parameters of the species tree and introgression events among branches precisely, including the strength and direction of introgression as well as species divergence times, introgression times, and effective population sizes. Current approximate methods are mostly unable to estimate these parameters or infer gene flow between sister lineages (*Jiao et al., 2021*; *Mirarab et al., 2021*).

Neotropical butterflies of the genus *Heliconius* have become a model system for understanding introgression (*Heliconius Genome Consortium, 2012*; *Martin et al., 2013*; *Kozak et al., 2021*; *Cicconardi et al., 2023*). Previous phylogenomic studies have demonstrated introgression among closely related species, but investigations of gene flow deeper among lineages were only partially successful: the various different methods yielded different phylogenies and introgression scenarios (*Kozak et al., 2021*; *Cicconardi et al., 2023*), perhaps due to methodological artifcts.

Recently, we demonstrated deep-level introgression and hybrid speciation in the erato-sara clade of *Heliconius* (*Edelman et al., 2019*; *Thawornwattana et al., 2022*). Here, we focus on the more complex melpomene crown group, or 'melpomene-silvaniform' group, which includes the melpomene-cydno-timareta clade, and 'silvaniform' species that are mostly Müllerian mimics of Ithomiini models, together with the related *Heliconius besckei* and *H. elevatus* (*Brown, 1981*). The melpomene-silvaniform species frequently hybridize today, and laboratory crosses demonstrate some interfertility across the entire group (*Mallet et al., 2007*). Thus, extensive gene flow is likely, making estimation of the true species phylogeny difficult. We also examine the overall species divergence and deeper introgression history of the entire genus. By using full-likelihood analysis of whole-genome data, we overcome limitations of approximate methods to obtain a robust estimate of the species tree, with major introgression events between branches quantified in terms of direction, strength, and timing. We use subsets of species to represent each major clade and answer specific questions of introgression, which helps to keep the computation manageable. Our species phylogeny and introgression history provide a more parsimonious explanation for evolution of key traits in *Heliconius* than those previously inferred using approximate methods (*Jay et al., 2018*; *Kozak et al., 2021*; *Cicconardi et al., 2023*). We find evidence of extensive autosomal gene flow across the melpomene-silvaniform group and show how trees based on the Z chromosome most likely represent the true species phylogeny. The so-called silvaniform species appear to be paraphyletic, contrary to previous findings based on approximate methods (*Zhang et al., 2016*; *Kozak et al., 2021*). As well as improving the understanding of diversification and gene flow in the genus *Heliconius*, we believe our approach provides useful pointers for studying species phylogeny with complex patterns of introgression in other taxa.

## Results

### Ancestral gene flow at the base of *Heliconius* phylogeny

We first establish phylogenetic relationships among six major clades of *Heliconius* using blocks of 200 loci (well-spaced short segments) across the genome, with each block spanning 1–3 Mb. We select a representative species from each major clade (*H. burneyi*, *H. doris*, *H. aoede*, *H. erato*, *H. sara*), three species from the melpomene-silvaniform group (*H. melpomene*, *H. besckei*, *H. numata*), and one outgroup species (*Eueides tales*) (*Supplementary file 1a–c*). Species tree search in each block was carried out under the MSC model using the Bayesian program BPP (see 'Methods'). Although each block was assumed to lack gene flow, different trees in each block across the genome are likely to result from introgression. Unlike a windowed concatenated tree approach, our MSC method takes

into account incomplete lineage sorting (ILS) and uncertainty in gene trees. Coding and noncoding loci were analyzed separately.

Two major patterns of species relationships were found at the base of the genus *Heliconius*: scenario 1 (*erato-early*), with the erato-sara clade diverging first, is supported by ~75% of genomic blocks, while scenario 2 (*aoede-early*), with *H. aoede* diverging first, is supported by ~20% of blocks (*Figure 1A*; see *Supplementary file 1d* for genome-level summaries and *Supplementary file 1e* for chromosome-level summaries). Using a different reference genome or more stringent filtering yielded similar fractions of the same species trees across the genome (*Figure 1—figure supplements 1 and 2*, *Supplementary file 1d–g*). In both scenarios, there is uncertainty concerning (i) the branching order of *H. doris* and *H. burneyi*, with *H. doris* diverging first being the most common relationship in both scenarios, and (ii) the branching order within the melpomene-silvaniform clade, with both *H. besckei + H. numata* and *H. melpomene + H. numata* being nearly equally common across the genome (*Figure 1B and C*). To focus on the deep divergences, we constructed simplified summaries of inferred species trees with *H. besckei, H. numata,* and *H. melpomene* grouped together ('BNM' in *Figure 1A*; also see *Figure 1—figure supplement 2* and *Supplementary file 1d and e*). Summaries of full species trees are provided in *Figure 1—figure supplement 1* and *Supplementary file 1f and g*; posterior probabilities of these local blockwise trees are generally low, with the median probability for each maximum a posteriori (MAP) tree <0.7, reflecting both limited information from only 200 loci and the challenge of resolving short branches in the species trees.

Scenarios 1 and 2 are related via ancestral gene flow. If scenario 1 (*Figure 1B*) represents the true species tree, gene flow between the erato-sara clade and the common ancestor of *H. doris*, *H. burneyi,* and the melpomene-silvaniform clade would lead to trees of scenario 2, with reduced estimated divergence time of the erato-sara clade. Similarly, if scenario 2 (*Figure 1C*) is the true tree, gene flow between the aoede clade and the common ancestor of *H. doris*, *H. burneyi,* and the melpomene-silvaniform clade would lead to trees of scenario 1, with reduced estimated divergence time of *H. aoede*. This expected reduction in divergence time as a result of introgression is not apparent in our species tree estimates, partly due to a short internal branch separating *H. aoede* and the erato-sara clade in both scenarios (*Figure 1B and C*). To assess which scenario fits the data better, we calculated the Bayes factor under the MSC model with introgression (MSC-I) implemented in ʙᴘᴘ, with *H. besckei* excluded to simplify the model (*Figure 1D and E*; 'etales-8spp' dataset in *Supplementary file 1a*). The Bayes factor provides mixed evidence, with different chromosomes either strongly supporting alternative scenarios or not significant at 1% level; however, the Z chromosome supports scenario 2 (*Supplementary file 1h*). In scenario 2, divergence of *H. aoede* was estimated to be older than that of the erato-sara clade in scenario 1 while root age estimates were comparable, further supporting the hypothesis that *H. aoede* most likely diverged before the erato-sara clade (*Figure 1—figure supplements 3 and 4*, *Supplementary file 1i and j*). This is because younger divergence may be explained by introgression whereas older divergence more likely represents true time of divergence. In scenario 2, introgression from the doris-burneyi-melpomene clade was estimated as unidirectional into *H. aoede* with a high probability (~75%), occurring shortly before the divergence of *H. doris* (*Figure 1—figure supplements 3 and 6*, *Supplementary file 1j*). Surprisingly, under scenario 1 we also find strong unidirectional introgression into the erato-sara clade with a high introgression probability (~65%) (*Figure 1—figure supplements 3 and 5*, *Supplementary file 1i*).

Two additional pieces of evidence support scenario 2. First, species trees of scenario 2 are most common in the Z chromosome (chr. 21) and tend to be more common in longer chromosomes, which have lower recombination rates per base pair (*Figure 1F*). Conversely, species trees of scenario 1 are more common in shorter chromosomes, which have higher recombination rates. If we assume that regions of low recombination tend to exhibit less introgression, with the Z chromosome usually most resistant to gene flow in *Heliconius* (*Appendix 1—figure 1*, *Figure 1—figure supplement 7*, *Supplementary file 1k*), as suggested by previous studies (*Edelman et al., 2019*; *Martin et al., 2019*), the association of species trees of scenario 2 with regions of low recombination suggests that scenario 2 more likely represents the true species relationships, whereas scenario 1 is a result of introgression. Second, we obtain a star tree whenever *H. erato* is assumed to diverge before *H. aoede* (scenario 1) under the MSC-with-migration (MSC-M or isolation-with-migration; IM) model of three species that allows for gene flow between two ingroup species since their divergence but does not allow gene flow with the outgroup species (*Appendix 1—figure 2*). Star trees are commonly observed when the

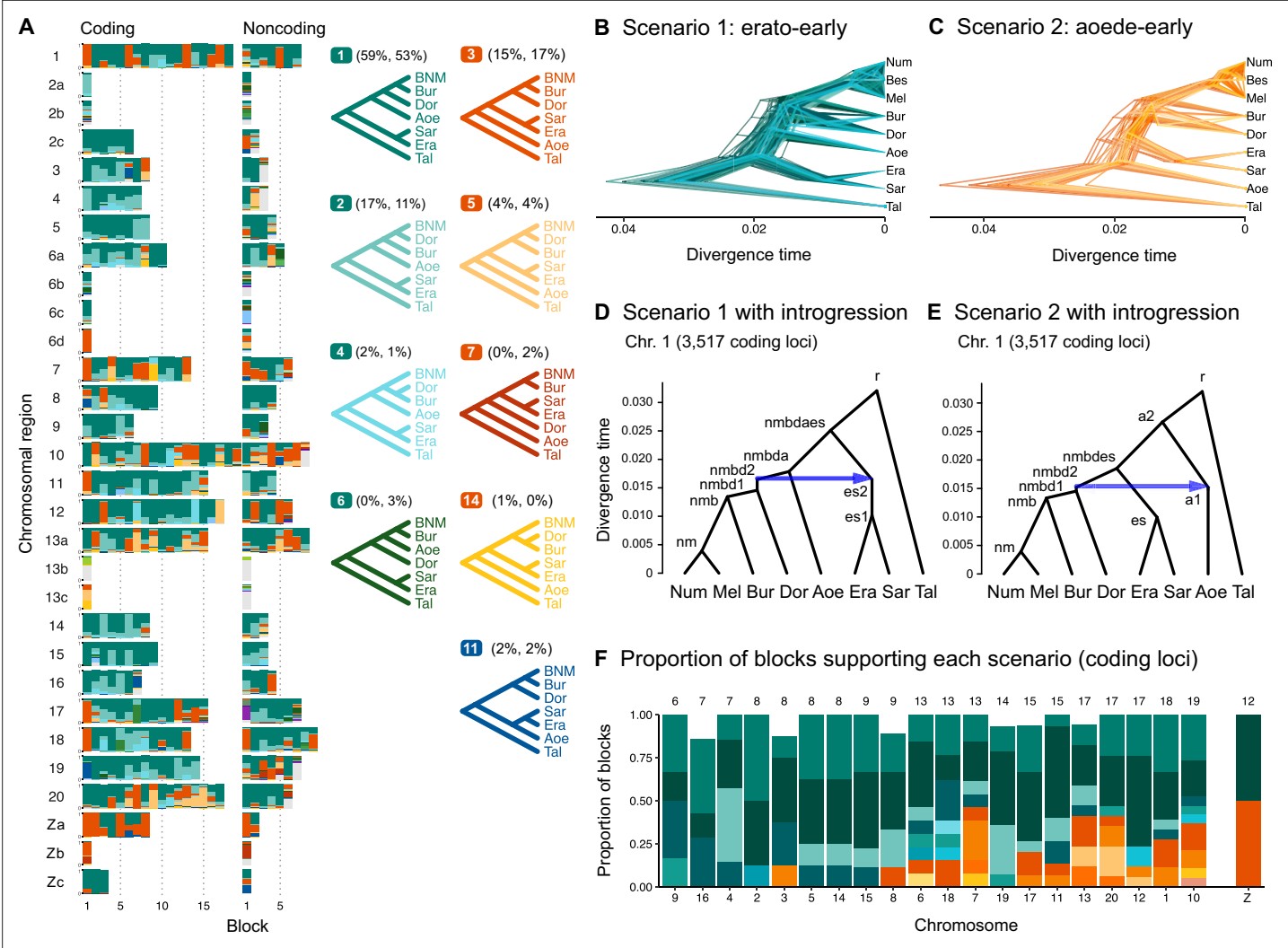

**Figure 1.** Ancestral gene flow at the base of *Heliconius*. (**A**) Posterior probabilities of species trees for major *Heliconius* clades inferred from BPP analysis of 200-locus blocks (each spanning 1–3 Mb) across the genome under the multispecies coalescent (MSC) model with no gene flow. The y-axis shows the posterior probability of species trees and ranges from 0 to 1. Colors correspond to the nine most common maximum a posteriori (MAP) trees, summarized by lumping species in the melpomene-silvaniform clade (Bes, Num, Mel) into a single tip (BNM); see *Figure 1—figure supplement 1* for full trees. Proportions of coding and noncoding blocks with each tree as a MAP tree are shown in parentheses. (**B, C**) Two scenarios of early divergence of *Heliconius*: (**B**) erato-early versus (**C**) aoede-early. Each tree is a MAP tree from a block having the MAP tree supporting one of these scenarios, with estimates of branch lengths (posterior means). (**D, E**) Phylogenetic and introgression histories estimated under an MSC model with introgression (MSC-I) corresponding to the two scenarios based on coding loci in chromosome 1 (3517 coding loci; see *Figure 1—figure supplements 5 and 6*). (**F**) Proportions of trees of scenarios 1 or 2 in each chromosome in order of increasing number of loci (used as a proxy for chromosome length). The Z chromosome (chr. 21) is placed at the right end. Number of blocks is shown on top of each bar. Tal: *Eueides tales*; Mel: *H. melpomene*; Bes: *H. besckei*; Num: *H. numata*; Bur: *H. burneyi*; Dor: *H. doris*; Aoe: *H. aoede*; Era: *H. erato*; Sar: *H. sara*.

The online version of this article includes the following figure supplement(s) for figure 1:

**Figure supplement 1.** Posterior estimates of species trees from four versions of the 'etales-9spp' dataset (see 'Methods'), using either *H. erato* (**A**, **C**) or *H. melpomene* (**B**, **D**) as a reference, and a read depth cutoff either 12 (**A**, **B**) or 20 (**C**, **D**).

**Figure supplement 2.** Posterior estimates of species trees from four versions of the 'etales-9spp' dataset (see 'Methods'), using either *H. erato* (**A**, **C**) or *H. melpomene* (**B**, **D**) as a reference, and a read depth cutoff either 12 (**A**, **B**) or 20 (**C**, **D**), with the clade containing three species (Bes, Num, Mel) in the melpomene-silvaniform clade collapsed as a single species.

**Figure supplement 3.** Posterior means and 95% highest posterior density (HPD) intervals of introgression probabilities ($\varphi$), population sizes ($\theta$), and divergence or introgression times ($\tau$) under scenario 1 and scenario 2 multispecies coalescent model introgression (MSC-I) model in *Figure 1D and E* obtained from BPP analysis of the 'etales-8spp' dataset (see *Supplementary file 1c* for the number of loci).

*Figure 1 continued on next page*

*Figure 1 continued*

**Figure supplement 4.** Posterior means and 95% HPD intervals of divergence times estimated under the multispecies coalescent with introgression (MSC-I) model of *Figure 1D* (erato-early, x-axis) versus estimates under the MSC-I model of *Figure 1E* (aoede-early, y-axis) using coding (**A**) or noncoding loci (**B**) of the 'etales-8spp' dataset.

**Figure supplement 5.** Posterior estimates of the species tree under the introgression model in *Figure 1D* (scenario 1: erato-early) obtained from BPP analysis of the 'etales-8spp' dataset for each chromosomal region.

**Figure supplement 6.** Posterior estimates of the species tree under the introgression model in *Figure 1E* (scenario 2: aoede-early) obtained from BPP analysis of the 'etales-8spp' dataset for each chromosomal region.

**Figure supplement 7.** Maximum likelihood estimates (MLEs) of all parameters in the MSC-M model as well as internal branch lengths ($\Delta\tau = \tau_0 - \tau_1$) by chromosomal region for all pairs of *Heliconius* species in the 'etales-9spp' dataset obtained from 3s analysis.

assumed bifurcating tree is incorrect (*Dalquen et al., 2017*). By contrast, assuming *H. aoede* diverges before *H. erato* (scenario 2) always leads to a bifurcating tree.

Four small inversion regions (~100–400 kb) had been identified previously to be differentially fixed between the melpomene group and the erato-sara clade (*Seixas et al., 2021*): 2b, 6b, 6c, 13b, and 21b. We were able to extract only a small number of loci (<100; see *Supplementary file 1c*) from each region. While species tree estimates are more uncertain (*Figure 1—figure supplements 1 and 2*, *Supplementary file 1d–g*), the 13b region (~360 kb with respect to *H. erato demophoon* reference) consistently shows a unique pattern in which *H. doris* and *H. burneyi* cluster with the erato-sara clade instead of the melpomene-silvaniform clade. This suggests ancient introgression of the inversion from the erato-sara clade into *H. doris* and *H. burneyi* (*Seixas et al., 2021*).

In conclusion, we detect some hitherto unrecognized introgression among the deepest branches within *Heliconius* sensu lato. The aoede-early scenario coupled with these deep introgression events may have led to some of the morphological and ecological evidence previously used in support of the erection the subgenera *Neruda* (for *H. aoede* and allies [*Turner, 1976*]) as well as *Laparus* (for *H. doris* [*Turner, 1968*]) that appeared to conflict with more recent molecular genetic data.

## Major introgression patterns in the melpomene-silvaniform clade

We next focus more closely on the melpomene-silvaniform clade (including *H. besckei, H. numata,* and *H. melpomene,* represented by 'BNM' in *Figure 1A*). This is one of the most phylogenetically difficult groups of *Heliconius* due to ongoing hybridization and extensive gene flow involving most members of the group (*Mallet et al., 2007*). Previous studies have inferred conflicting introgression scenarios in this clade (*Martin et al., 2013*; *Zhang et al., 2016*; *Jay et al., 2018*; *Edelman et al., 2019*; *Cicconardi et al., 2023*). We compiled a multilocus dataset from high-quality genomic data comprising eight (out of 15) species representing all major lineages within the clade: *H. melpomene, H. cydno, H. timareta, H. besckei, H. numata, H. hecale, H. elevatus,* and *H. pardalinus* (*Supplementary file 1l*). Our analysis below confirms widespread introgression within this clade.

We identify four major species relationships from blockwise estimates of species trees under the MSC model without gene flow (*Figure 2A*, *Figure 2—figure supplement 1*, *Supplementary file 1m and n*): (a) autosome-majority (trees 1–3), (b) autosome-variant (trees 5–7), (c) the Z chromosome (chr. 21; tree 4), and (d) chromosome 15 inversion region (15b; tree 24). The pattern is highly similar between coding and noncoding loci. The first three relationships (trees 1–9) account for >90% of the blocks, with a well-supported pardalinus-hecale clade ((*H. pardalinus, H. elevatus*), *H. hecale*) and a cydno-melpomene clade ((*H. timareta, H. cydno*), *H. melpomene*). They differ in the position of *H. numata* and in the relationships among the three species in the cydno-melpomene clade. We first focus on the three scenarios relating to different positions of *H. numata*: (a) *H. numata* sister to the pardalinus-hecale clade + the cydno-melpomene clade, (b) *H. numata* sister to the pardalinus-hecale clade, and (c) *H. numata* sister to *H. besckei*. The Z-chromosome tree (i.e. tree 4 in scenario c) is the MAP tree with a high posterior probability in almost all blocks of the Z chromosome (median probability of 1; *Figure 2A*, *Supplementary file 1m*), with *H. besckei* + *H. numata* diverging first, followed by a split between the pardalinus-hecale clade and the cydno-melpomene clade. A similar distinction between the autosome-majority trees and the Z-chromosome tree was also found by *Zhang et al., 2016*. All four scenarios a–d confirm paraphyly of the silvaniform species (*H. besckei, H. numata,* and the pardalinus-hecale clade), consistent with some recent phylogenomic studies (*Zhang et al., 2016*; *Massardo et al., 2020*; *Cicconardi et al., 2023*). Monophyly of the silvaniforms was suggested in

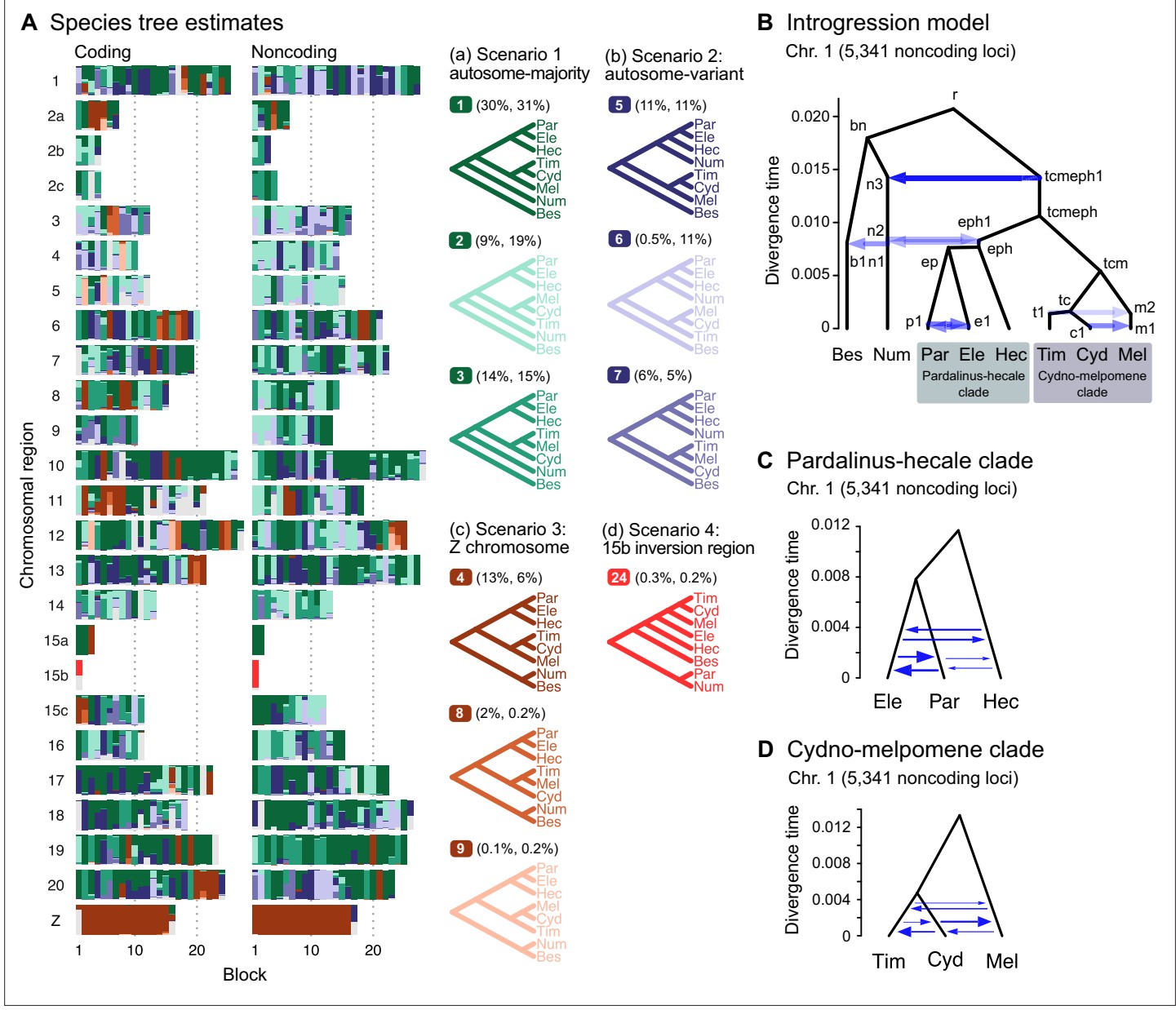

**Figure 2.** Major introgression events in the melpomene-silvaniform clade. (**A**) Blockwise estimates of species trees of the melpomene-silvaniform clade inferred from 200-locus blocks across the genome under the multispecies coalescent (MSC) model with no gene flow using BPP (see *Supplementary file 1l* for data, *Supplementary file 1m and n* and *Figure 2—figure supplement 1* for full results). Maximum a posteriori (MAP) trees are labeled in decreasing order of frequency among blocks. Proportions of coding and noncoding blocks with each tree as a MAP tree are shown in parentheses. (**B**) The multispecies coalescent with introgression (MSC-I) model events that can explain the three major groups of trees in (**A**). Branch lengths are based on posterior means of divergence/introgression times estimated from 5341 noncoding loci on chromosome 1 (*Supplementary file 1q*). Each internal node is given a label, which is used to refer to a population above the node, for example, the population between nodes r and bn is referred to as branch bn. Each horizontal arrow represents a unidirectional introgression event, for example, the arrow from tcmeph1 to n3 represents tcmeph→Num introgression at time $\tau_{\text{tcmeph1}} = \tau_{\text{n3}}$ with probability $\varphi_{\text{tcmeph1}\to\text{n3}}$. (**C**) Continuous migration (IM) model for the pardalinus-hecale clade, allowing bidirectional gene flow among the three species. (**D**) Multispecies coalescent with migration (MSC-M) model for the cydno-melpomene clade. For (**C**) and (**D**), branch lengths are based on estimates from noncoding loci in chromosome 1 (*Figure 2—figure supplement 6*, *Supplementary file 1s–u*), and arrow sizes are proportional to estimated migration rate (*M = Nm*). Bes: *H. besckei*; Num: *H. numata*; Par: *H. pardalinus*; Ele: *H. elevatus*; Hec: *H. hecale*; Tim: *H. timareta*; Cyd: *H. cydno*; Mel: *H. melpomene*.

The online version of this article includes the following figure supplement(s) for figure 2:

**Figure supplement 1.** Posterior estimates of species trees of the melpomene-silvaniform clade inferred from 200-locus blocks across the genome from the BPP multispecies coalescent (MSC) analysis with no gene flow of the 'hmelv25-res' dataset.

*Figure 2 continued on next page*

*Figure 2 continued*

**Figure supplement 2.** Maximum likelihood estimates (MLEs) of all parameters in the multispecies coalescent with migration (MSC-M) model as well as internal branch lengths ($\Delta\tau = \tau_0 - \tau_1$) by chromosomal region from species pairs in the 'hmelv25-all' dataset obtained from 3s analysis using (**A**) *H. aoede* or (**B**) *H. erato* as an outgroup.

**Figure supplement 3.** Estimated introgression history.

**Figure supplement 4.** Posterior means and 95% HPD intervals of (**A**) introgression probabilities, (**B**) divergence or introgression times ($\tau$), and (**C**) population sizes ($\theta$) under the multispecies coalescent with introgression (MSC-I) model of *Figure 2B* using coding (blue) and noncoding data (yellow).

**Figure supplement 5.** Posterior means and 95% HPD intervals of species divergence and introgression times ($\tau$) obtained from coding loci (*y*-axis) versus noncoding loci (*x*-axis) under the multispecies coalescent with introgression (MSC-I) models of *Figure 2B* (for chromosomal regions other than 15b) and of (**D**) (for 15b).

**Figure supplement 6.** Posterior means and 95% HPD intervals of migration rates (M), divergence times ($\tau$), and population sizes ($\theta$) under the multispecies coalescent with migration (MSC-M) model in three species obtained using BPP.

concatenation/sliding-window analysis (*Heliconius Genome Consortium, 2012*; *Kozak et al., 2015*; *Kozak et al., 2021*; *Zhang et al., 2021*), but this conclusion may suffer from a failure to account for deep coalescence (*Edwards et al., 2016*).

Our species tree search under the MSC (*Figures 1A and 2A*) accounts for ILS but does not account for gene flow. To approximate a fuller introgression history of this group, we construct a species tree model with introgression that can explain the four scenarios above. We use estimates of migration rates between each pair of species under the MSC-with-migration (MSC-M or IM) model of species triplets (3s analysis) to inform placement of introgression edges (*Appendix 1—figure 2*, *Figure 2—figure supplement 2*). Our proposed model has six pairs of bidirectional introgression events (*Figure 2B*). We use the Z-chromosome tree (tree 4) as the backbone that most likely represents the true species tree, while conflicting trees result from historical introgression. Tree 4 is largely limited to the Z chromosome, which appears more resistant to gene flow in *Heliconius* (*Zhang et al., 2016*; *Van Belleghem et al., 2018*; *Martin et al., 2019*; *Massardo et al., 2020*; *Thawornwattana et al., 2022*). Consistent with this interpretation, we find no evidence of gene flow in the Z chromosome and high prevalence of gene flow on the autosomes based on the 3s analysis (*Appendix 1—figure 2*, *Figure 2—figure supplement 2*, *Supplementary file 1o and p*). Thus, we include two introgression events (between nodes n3–tcmeph1 and n2–eph1 in *Figure 2B*) to explain alternative positions of *H. numata* in the autosomes (scenarios a and b). Next, to explain the secondary source of genealogical variation within the cydno-melpomene clades (i.e. among trees 1–3, trees 4/8/9 and trees 5–7), we add two further introgression events between *H. melpomene* and *H. cydno* (m1–c1), and between *H. melpomene* and *H. timareta* (m2–t1). We do not model *H. cydno–H. timareta* introgression (these species are allopatric). We also include introgression between *H. besckei* and *H. numata* (b1–n1) to relate the autosome trees to the Z-chromosome tree. Finally, sister species *H. pardalinus* and *H. elevatus* hybridize today in sympatric populations (*Rosser et al., 2019*), so we allow introgression between them (p1–e1). According to the 3s analysis, the rates of gene flow in this pair are among the highest (*Figure 2—figure supplement 2*). This sister-species introgression does not alter species trees (*Figure 2A*) because it does not change the topology.

The resultant MSC-I model is used to estimate species divergence times and effective population sizes for extant and ancestral species, and the intensity, timing, and direction of introgression. Consistent with scenario c representing the true species tree, we find least introgression on the Z chromosome (*Figure 2—figure supplements 3 and 4*, *Supplementary file 1q and r*). On the autosomes, there is substantial introgression from the pardalinus-hecale + cydno-melpomene clade into *H. numata*, and to a lesser extent, between *H. numata* and the pardalinus-hecale clade. These patterns match well with the frequencies of the two main autosomal relationships (scenarios a and b in *Figure 2*). Within the cydno-melpomene clade, introgression is predominantly unidirectional from *H. cydno* and *H. timareta* into *H. melpomene*. The *H. pardalinus–H. elevatus* pair shows ongoing extensive but variable introgression across the genome, with the introgression time estimated to be zero. See Appendix 1, section 'Major introgression patterns in the melpomene-silvaniform clade inferred using 3s and BPP' for more details.

The age of the melpomene-silvaniform clade ($\tau_r$) is estimated to be ~0.020 substitutions per site based on noncoding data (*Figure 2B*, *Figure 2—figure supplement 4*, *Supplementary file 1q*). This

translates to ~1.7 (CI: 0.9, 3.8) million years ago (Ma), assuming a neutral mutation rate of $2.9 \times 10^{-9}$ per site per generation (95% CI: $1.3 \times 10^{-9}$, $5.5 \times 10^{-9}$) and four generations per year (***Keightley et al., 2015***). This is not very different from a previous estimate of 3.7 (CI: 3.2, 4.3) Ma based on molecular clock dating (***Kozak et al., 2015***), which ignores ancestral polymorphism and is therefore expected to overestimate divergence time. Overall, our estimates of species divergence time tend to be precise and highly similar across the genome (***Figure 2—figure supplements 3 and 4***). The posterior means from coding and noncoding loci are strongly correlated, with $\tau_C \approx b\tau_{NC}$ where $b$ varies between 0.4 and 0.6 ($r^2 > 0.95$) in most chromosomal regions (***Figure 2—figure supplement 5***). The scale factor of $b <$ 1 can be explained by purifying selection removing deleterious nonsynonymous mutations in coding regions of the genome (***Shi and Yang, 2018***). Present-day and ancestral population sizes ($\theta$) are of the order of 0.01 (***Figure 2—figure supplement 4B***, ***Supplementary file 1q and r***). For inbred individuals (chosen for sequencing to facilitate genome assembly) among our genomic data (*H. melpomene*, *H. timareta*, *H. numata,* and *H. pardalinus*; see ***Supplementary file 1a***), $\theta$ estimates vary among chromosomes by orders of magnitude, with the inbred genome of *H. melpomene* having the lowest population size of ~0.002–0.004 on average. Adding more individuals should help stabilize estimates of $\theta$, but should not affect estimates of age or introgression rates.

The introgression model of ***Figure 2B*** assumes that gene flow occurs in single pulses. This may be unrealistic if gene flow is ongoing. We thus employ the MSC-M model implemented in BPP to estimate migration rates ($M = Nm$) between all pairs of species in each of the pardalinus-hecale and cydno-melpomene clades (***Figure 2C and D***). The MSC-M model assumes continuous gene flow since lineage divergence at the rate of $M$ migrants per generation (***Flouri et al., 2023***). The results concur with the introgression pulse model in suggesting high gene flow between *H. pardalinus* and *H. elevatus* (***Figure 2—figure supplement 6A and B***, ***Supplementary file 1s***). There is also evidence of gene flow between *H. hecale* and *H. pardalinus*/*H. elevatus* at lower levels, with $M < 0.1$ in most chromosomes (***Figure 2—figure supplement 6A***, ***Supplementary file 1s***). Allowing for continuous gene flow as well as gene flow involving *H. hecale*, we obtain slightly older estimates of both species divergence times (between *H. pardalinus* and *H. elevatus*, and between *H. hecale* and *H. pardalinus* + *H. elevatus*) (***Figure 2—figure supplement 6A***, ***Supplementary file 1s***), than under the single-pulse introgression model (***Supplementary file 1q and r***). The cydno-melpomene clade shows a similar pattern of older divergence times with substantial gene flow between all three species although at smaller magnitudes ($M$ ~0.01–0.1) (***Figure 2—figure supplement 6C***, ***Supplementary file 1u***). For more discussion of parameter estimates, see 'Appendix 1,' section 'MSC-M model for pardalinus-hecale and cydno-melpomene clades.' We conclude that a model with continuous gene flow involving all three species may better describe the history of both the pardalinus-hecale clade and the cydno-melpomene clade.

In summary, we have identified substantial gene flow within the cydno-melpomene and pardalinus-hecale clades based on both pulse introgression (MSC-I) and continuous migration (MSC-M) models (***Figure 2C and D,***, ***Figure 2—figure supplement 6***, ***Supplementary file 1q–u***). Earlier genomic studies failed to quantify the intensity of gene flow (introgression probability or migration rate) or infer direction and timing of gene flow. Gene flow within the cydno-melpomene clade has been extensively studied at population/subspecies levels and at specific loci involved in wing pattern mimicry (***Bull et al., 2006***; ***Pardo-Diaz et al., 2012***; ***Kronforst et al., 2013***; ***Martin et al., 2013***; ***Wallbank et al., 2016***; ***Enciso-Romero et al., 2017***; ***Martin et al., 2019***), but gene flow involving other species has received less attention (***Heliconius Genome Consortium, 2012***; ***Wallbank et al., 2016***; ***Zhang et al., 2016***; ***Jay et al., 2018***).

## Complex introgression in the 15b inversion region (P locus)

A polymorphic series of tandem inversions on chromosome 15 is involved in switching mimicry color pattern in *H. numata* (***Jay et al., 2018***; ***Jay et al., 2022***). The first inversion, $P_1$ (~400 kb), is in the 15b region (also called the *P* locus), and is fixed in *H. pardalinus* and retained as a polymorphism in *H. numata* (***Joron et al., 2011***; ***Le Poul et al., 2014***; ***Jay et al., 2018***). Multiple introgression events are necessary to make the 15b tree (***Figure 2A***, scenario d, tree 24) compatible with either the Z chromosome tree or the autosomal trees (***Figure 2A***, scenarios a–c), suggesting a much more complex introgression history of this region than in the rest of the genome. This inversion contains the known wing patterning locus *cortex* (***Jay et al., 2022***), where it is maintained as a balanced polymorphism

by natural selection (*Joron et al., 2006*; *Nadeau et al., 2016*; *Van Belleghem et al., 2017*). Another feature of this 15b region is that among the species without the inversion, the cydno-melpomene clade clusters with *H. elevatus* and is nested within the pardalinus-hecale clade (without *H. pardalinus*). This is contrary to the expectation based on the topologies in the rest of the genome (*Figure 2A*, scenarios a–c) that the cydno-melpomene clade would be sister to the pardalinus-hecale clade (without *H. pardalinus*). One explanation for this pattern is that introgression occurred between the common ancestor of the cydno-melpomene clade and either *H. elevatus* or the common ancestor of *H. elevatus* and *H. pardalinus* together with a total replacement of the non-inverted 15b in *H. pardalinus* by the $P_1$ inversion from *H. numata* (*Jay et al., 2018*). We confirm and quantify this introgression below.

Using data from additional species ('silv_chr15' dataset in *Supplementary file 1a*; see 'Methods'), we obtain a better resolution of species relationships along chromosome 15, although with some uncertainty within the inversion region due to small numbers of loci (*Figure 3A*, *Supplementary file 1v*). This analysis of independent data agrees with the Z chromosome tree (tree 24 in *Figure 2A*) and with (unrooted) trees obtained from concatenation analysis by *Jay et al., 2018* (their Figures 2 and S1) and *Jay et al., 2021* (their Figure S4), where *H. numata* with the inversion groups with *H. pardalinus* while *H. numata* without the inversion groups with its sister species, *H. ismenius* (*Figure 3A*, red trees). Outside the inversion region, *H. numata* with both inversion genotypes groups with *H. ismenius* as expected (*Figure 3A*, blue trees). Although this conclusion assumes that *H. numata* and *H. ismenius* are sister species while *H. ismenius* is not included in our species tree analysis of the melpomene-silvaniform clade (*Figure 2*), this sister relationship agrees with previous genomic studies of the autosomes and the sex chromosome (*Zhang et al., 2016*; *Jay et al., 2021*; *Cicconardi et al., 2023*; *Rougemont et al., 2023*).

Given that *H. numata* is an early-diverging lineage of the melpomene-silvaniform clade (*Figure 2B*) and is polymorphic for $P_1$ over large parts of its geographic distribution while *H. pardalinus* is fixed for this inversion (*Joron et al., 2011*; *Jay et al., 2022*), one parsimonious explanation is that the inversion originated either in *H. numata* after diverging from *H. ismenius*, or before the *H. numata*–*H. ismenius* split but was subsequently lost in *H. ismenius*, followed by introgression which introduced the inversion from *H. numata* into *H. pardalinus*, either before or after its divergence from *H. elevatus* (*Figure 3B*). If the introgression occurred before *H. pardalinus*–*H. elevatus* divergence, the lack of the inversion in *H. elevatus* can be explained by another introgression from the cydno-melpomene clade into *H. elevatus*, completely replacing the inversion with the original orientation (*Figure 3B*). This introgression route is reported in previous genomic studies, including at a different locus (*optix* gene, also involved in color pattern) in chromosome 18 as well as in the 15b region of chromosome 15 (*Heliconius Genome Consortium, 2012*; *Wallbank et al., 2016*). Under this scenario, we might expect the 15b region to be less diverse in *H. pardalinus* (recipient of $P_1$) than in *H. numata* (donor of $P_1$), with the magnitude depending on the duration between the origin of $P_1$ and introgression into *H. pardalinus*. However, we do not see this reduced heterozygosity in our data (*Figure 3—figure supplement 1*), suggesting that the transfer likely occurred early, shortly after the formation of the inversion in *H. numata*, or shortly after the *H. numata*–*H. ismenius* split if the inversion originated earlier. This scenario is further supported by the 15b tree having similar times of divergence between *H. ismenius* and *H. numata* without the inversion, and between *H. pardalinus* and *H. numata* with the inversion (*Figure 3—figure supplement 2*). The topology of the 15b tree (*Figures 2A and 3A*) also indicates that the first split is between species with the inversion (*H. numata* and *H. pardalinus*) and those without the inversion. This suggests another possibility: inversion polymorphism could have existed earlier in the history of the melpomene-silvaniform clade but was subsequently lost in most species (*Figure 3C*). In this scenario, the ancestral polymorphism is maintained in *H. numata* while the inversion is fixed in *H. pardalinus* but is lost in other species. Introgression between *H. numata* and *H. pardalinus* is not required but could still occur.

To reconcile the introgression history of 15b with the overall species tree, we add three additional bidirectional introgression events onto the main model in *Figure 2B* and assess their plausibility using BPP. We allow for (i) bidirectional introgression between the cydno-melpomene clade into *H. elevatus*, (ii) bidirectional introgression between *H. numata* and *H. pardalinus*, and (iii) introgression between *H. besckei* and the common ancestor of the cydno-melpomene + pardalinus-hecale clade (to account for *H. besckei* being clustered with other species that do not have the inversion; see *Figure 2A*). We consider five models (m1–m5) differing in the placements of introgression events (i) and (ii) either

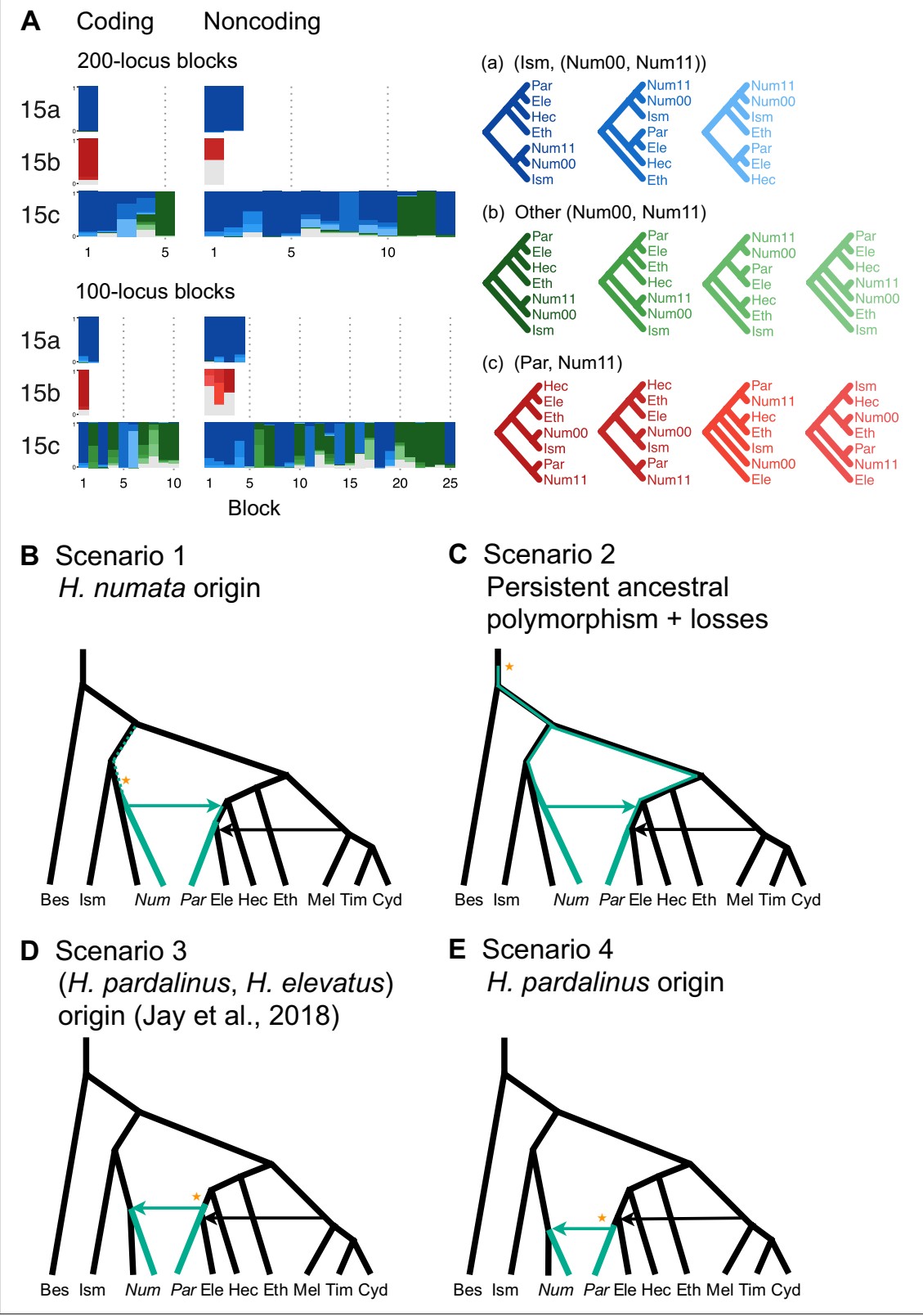

**Figure 3.** Introgression history of the chromosome 15 inversion region (15b). (**A**) Blockwise estimates of species trees for the inversion (15b) and the remnant flanking regions (15a and 15c) of chromosome 15. Trees were inferred from 200-locus blocks and 100-locus blocks under the multispecies coalescent (MSC) model without gene flow using BPP (***Supplementary file 1v***). Tree legends are grouped by whether *H. numata* clusters with *H. ismenius* (blue), or *H. numata* with P₁ inversion (Num11) clusters with *H. pardalinus* (red), or other relationships (green). (**B–E**) Four possible scenarios of the origin

*Figure 3 continued on next page*

*Figure 3 continued*

and introgression route of the P₁ inversion. Star indicates the origin of the P₁ inversion. Green lineages have the inversion, and green arrows indicates introgression of the inversion. Ism: *H. ismenius*; Num00: *H. numata* homozygous uninverted 15b (*H. n. laura* and *H. n. silvana*); Num11: *H. numata* homozygous for inversion P₁ (*H. n. bicoloratus*); Eth: *H. ethilla*. See *Figure 2* legend for codes for other species. For the direction of melpomene-cydno clade→*H. elevatus*, see *Figure 3—figure supplement 3*.

The online version of this article includes the following figure supplement(s) for figure 3:

**Figure supplement 1.** Mean heterozygosity per site of the 15b inversion region and flanking regions (15a and 15c) of *H. numata* (Num) and *H. pardalinus* (Par) individuals.

**Figure supplement 2.** Posterior estimates of divergence time for (Ism, Num00) and (Par, Num11) in the 15b region.

**Figure supplement 3.** Estimated introgression history under five introgression models (m1–m5) of the 15b region.

before or after the *H. pardalinus–H elevatus* split (*Figure 3—figure supplement 3*). Our results best support unidirectional introgression from *H. numata* into the common ancestor of *H. pardalinus* and *H. elevatus*, and from the common ancestor of the cydno-melpomene clade into *H. elevatus* shortly after its divergence from *H. pardalinus* (*Figure 3—figure supplement 3*, model m3). In other scenarios, estimated introgression times tend to collapse onto the *H. pardalinus–H elevatus* divergence time, suggesting that the introgression events were likely misplaced (*Figure 3—figure supplement 3*, *Supplementary file 1w*). Our finding that divergence of *H. elevatus* and introgression from the cydno-melpomene clade occurred almost simultaneously provides evidence for a hybrid speciation origin of *H. elevatus* resulting from introgression between *H. pardalinus* and the common ancestor of the cydno-melpomene clade (*Rosser et al., 2019*).

In an alternative scenario proposed by *Jay et al., 2018*, the inversion first originated in the common ancestor of *H. pardalinus* and *H elevatus*, and subsequently introgressed into some subspecies of *H. numata*, while the inversion in *H. elevatus* was completely replaced by introgression from the cydno-melpomene clade (*Figure 3D*). They used sliding-window gene tree topologies to support introgression of the inversion from *H. pardalinus* to *H. numata* shortly after its formation in the common ancestor of *H. pardalinus* and *H elevatus* (their Figures 4 and S4). By including *H. ismenius* and *H. elevatus*, sister species of *H. numata* and *H. pardalinus,* respectively, different directions of introgression should lead to different gene tree topologies. Clustering of (*H. numata* with the inversion, *H. pardalinus*) with *H. numata* without the inversion would suggest the introgression is *H. numata* → *H. pardalinus* while clustering of (*H. numata* with the inversion, *H. pardalinus*) with *H. elevatus* would suggest *H. pardalinus* → *H. numata* introgression. However, tree topologies supporting each direction of introgression were almost equally common within the inversion region, particularly in the first half of the inversion, undermining this argument. With high levels of ILS and introgression in the group, estimated gene trees need not reflect the true species relationships. Another variant of this scenario is that the inversion originated in *H. pardalinus* after its divergence from *H. elevatus* and was introgressed into some subspecies of *H. numata* (*Figure 3E*). We consider this scenario unlikely because the inversion appears to originated long before the *H. pardalinus–H. elevatus* split given deep divergence of lineages with and without the inversion (see 15b trees in *Figures 2A and 3A*).

## Discussion
### Approaches for estimating species phylogeny with introgression from whole-genome sequence data: Advantages and limitations

Our full-likelihood MSC approach yields several improvements over concatenation and other approximate methods for inferring species trees and introgression. First, we account for ILS due to coalescent fluctuations, so variation in inferred genealogical histories can be more firmly attributed to variation in gene flow across the genome. Second, we analyze the sequence data directly, rather than using estimated gene trees as data. This utilizes information in branch lengths of the gene trees while accommodating their uncertainty. We retain heterozygous sites in the alignments as unphased diploid sequences from each individual without collapsing them into a single randomly phased nucleotide as is common in other phylogenomic studies. Our approach allows multiple individuals per species to be included, leading to improved estimation of introgression parameters. Third, MSC models with pulse introgression (MSC-I) or continuous migration (MSC-M) allow direct estimation of key features of the

direction, intensity, and timing of gene flow, and are applicable to gene flow between sister species. Widely used summary statistics such as $D$ and $f_d$ do not estimate these parameters and cannot detect gene flow between sister lineages. Previous analyses using sliding-window concatenation or based on estimated gene trees were therefore liable to be less successful for estimating branching order in the melpomene-silvaniform clade, and because they may be misled by rapid speciation events coupled with extensive gene flow (*Edelman et al., 2019*; *Massardo et al., 2020*; *Kozak et al., 2021*).

There are several limitations in our approach. First, in the exploratory analysis of genomic blocks using MSC without gene flow (e.g. *Figures 1A and 2A*), we assume that one of the main topologies represents the true species tree and is related to other topologies via introgression. In general, it can be nontrivial to decide which topology most likely represents the true species tree, and how to add introgression edges onto the assumed species tree to explain other topologies. Here, we employ several strategies including a recombination-rate argument (*Figure 1F*), reduced introgression in the sex chromosome (*Figure 2A*), estimates of pairwise migration rates under an explicit model of gene flow (*Appendix 1—figures 1 and 2*), and using additional evidence from geographic distributions (*Rosser et al., 2012*) and records of natural hybrids (*Mallet et al., 2007*). These strategies could fail in taxa with more limited biological or distribution information, or when the sex chromosome (if available) is less unreliable as to species topology. Ideally, one would like to be able to estimate species phylogeny and introgression events simultaneously as part of a single unsupervised analysis, but this task remains computationally challenging. Software such as PhyloNet (*Wen and Nakhleh, 2018*) and starBEAST (*Zhang et al., 2018*) implement Bayesian inference of MSC with introgression but they are usually limited to small datasets of no more than 100 loci. Heuristics based on summary statistics may attempt to place introgression edges onto a species phylogeny estimated on the assumption of no gene flow (*Malinsky et al., 2018*; *Suvorov et al., 2022*; *Cicconardi et al., 2023*). However, these approaches do not make full use of sequence data and can be misled by incorrect initial species trees. Furthermore, they cannot convincingly infer direction of gene flow or gene flow between sister species.

Second, although our approach is more powerful than approximate methods, it is also more computationally intensive and does not scale well to analyses of large numbers of species (>20, say). In *Heliconius*, there are 47 species in six major clades. Our strategy has therefore been to analyze subsets of species representing major clades or representative species within each clade. We then combine the results into a larger phylogeny. One caveat is that not all species get explored at the same time, and some introgression events may be missed. Lastly, the MSC makes simplifying modeling assumptions such as neutral evolution, constant substitution rates on all branches, no recombination within each locus, and free recombination among loci, the implications of which have been discussed previously (*Burgess and Yang, 2008*; *Thawornwattana et al., 2022*). We note that although the MSC model assumes constant rate neutral evolution, estimated introgression probabilities ($\varphi$) are a combination of both actual introgression and subsequent positive and negative selection on introgressed loci, the extent of which is modulated by local recombination rate (*Petry, 1983*; *Barton and Bengtsson, 1986*).

In summary, we first explore major species relationships across genome blocks using a full-likelihood MSC approach. We then identify introgression events that explain the different local species trees by proposing introgression edges on a background species topology, with additional evidence from explicit models of gene flow. In addition, geographic distributions and the prevalence of natural hybrids can be employed to help with placement of introgression edges in a phylogeny. This process can result in several competing introgression models. We estimate parameters under each model using sequence data, evaluate the model fit, and perform model comparison. By breaking the problem into manageable parts in this way, our approach is computationally tractable.

## An updated phylogeny of *Heliconius*

We have clarified phylogenetic relationships among major clades within *Heliconius* and quantified major introgression events, including introgression between sister species (which does not change tree topology). We summarize our three key findings in *Figure 4*. First, *H. aoede* is most likely the earliest-branching lineage of *Heliconius*, and an ancient introgression event led to its apparently closer relationship with the doris-burneyi-melpomene clade (*Figure 1*). We discuss an implication of this placement below. However, we emphasize the considerable level of uncertainty that remains,

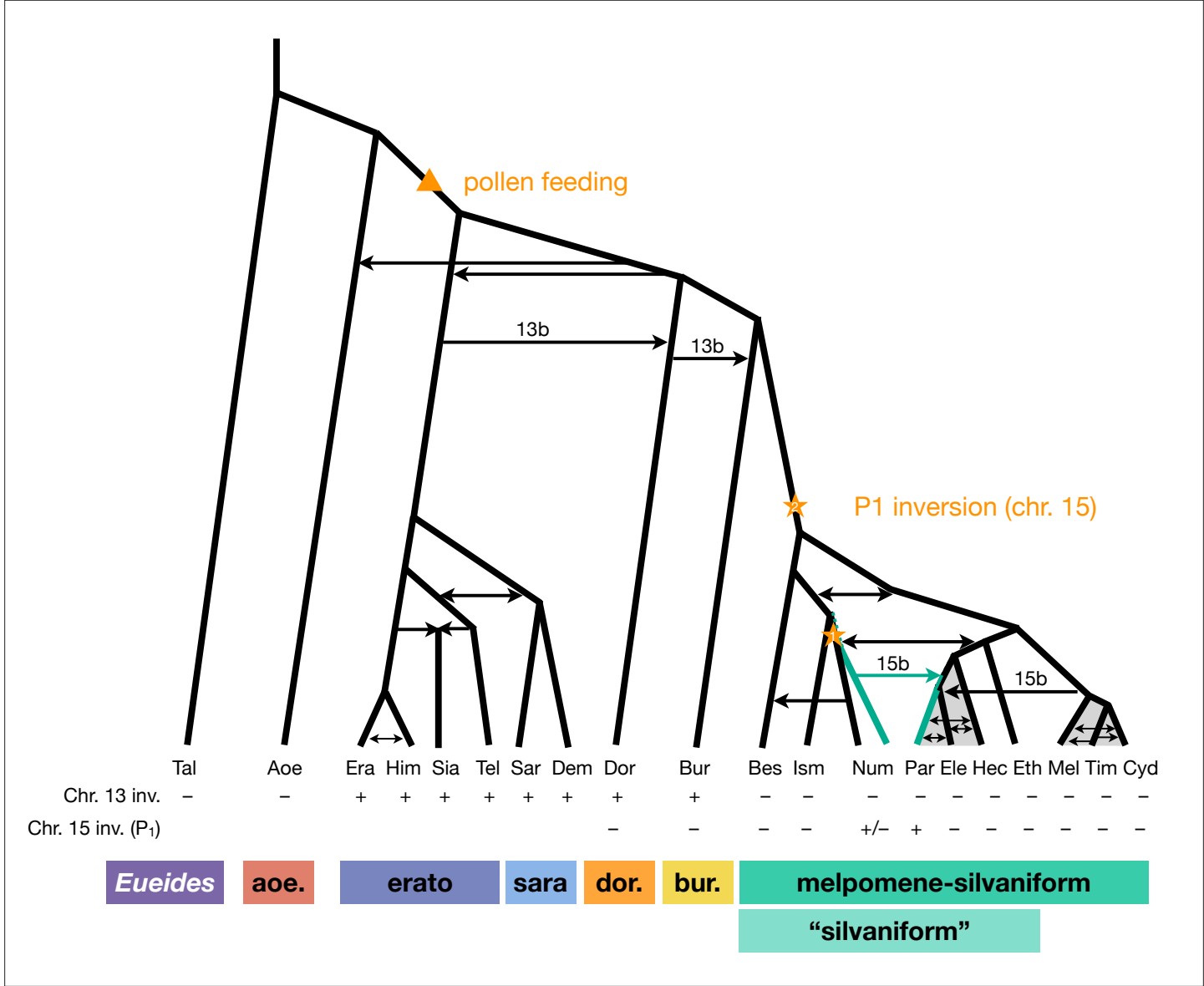

**Figure 4.** Revised phylogeny and introgression history of *Heliconius*. Phylogeny of the erato-sara clade was previously estimated using a similar approach (***Thawornwattana et al., 2022***). Arrows represent introgression events. Introgression of chromosomal inversions are region-specific, indicated by a label (13b or 15b) above the arrow. Status of inversions 13b and 15b in each species is indicated in the two rows at the bottom, with + indicating inversion, – indicating standard orientation, and +/– inversion polymorphism. Gray shading represents a period of continuous gene flow. Triangle represents the origin of pollen feeding near the base of *Heliconius*. Star indicates a possible origin of the P₁ inversion on chromosome 15, and green branches indicate lineages with the inversion (polymorphic in Num, fixed in Par). Him: *H. himera*; Sia: *H. hecalesia*; Tel: *H. telesiphe*; Dem: *H. demeter*; see ***Figures 1–3*** legends for other species codes.

and having more high-quality genome data from *H. aoede* and its sibling species may help improve the phylogenetic resolution. Second, within the melpomene-silvaniform clade, we obtained a robust pattern in which the common ancestor of *H. besckei*, *H. numata,* and *H. ismenius* first split off from the rest, rendering the silvaniform species paraphyletic (***Figure 2***). This is contrary to a common belief that the silvaniforms are monophyletic (***Heliconius Genome Consortium, 2012***; ***Kozak et al., 2015***; ***Jay et al., 2021***; ***Kozak et al., 2021***; ***Zhang et al., 2021***). Third, the P₁ inversion on chromosome 15 involved in wing pattern mimicry likely introgressed from *H. numata* into *H. pardalinus* based on species tree topologies (***Figures 2 and 3***) and direct modeling of introgression (***Figure 3—figure supplement 3***), although where and when it originated remain uncertain.

Overall, we conclude that the $P_1$ inversion likely arose in *H. numata* or its close ancestor and became fixed in *H. pardalinus* via an introgression replacement event from the former species (*Figure 3B*), although a deeper origin followed by persistent ancestral polymorphism (*Figure 3C*) remains a possibility.

*Heliconius* is an example of recent and rapid diversification with extensive gene flow occurring between closely related species as well as between more divergent lineages throughout its history at varying degrees. Since many species pairs of *Heliconius* show high degrees of sympatry today, there are plenty of opportunities for low levels of gene flow to occur whenever their ranges overlap (*Rosser et al., 2015*). This fact is evident in our estimates of pairwise migration rates among species in the melpomene-silvaniform clade (*Appendix 1—figures 1 and 2*). There are many documented natural hybrids (*Mallet et al., 2007*) and there is evidence for introgression between particular pairs of subspecies from genomic studies, most of which involve wing patterning loci (*Wallbank et al., 2016*; *Enciso-Romero et al., 2017*; *Morris et al., 2020*; *Rougemont et al., 2023*). We did not attempt to incorporate all possible introgression signals into our phylogeny but instead focused on major introgression events that explain major patterns of species tree variation across the genome using a representative subset of species. Thus our phylogeny in *Figure 4* should be viewed as a simplified version of reality. One future direction is to model a low level of admixture among all lineages.

## Implications for the evolution of pollen feeding

To illustrate how our updated phylogeny (*Figure 4*) can provide new insights into the evolution of *Heliconius*, we discuss an implication of our results for the evolution of pollen collection and pollen-feeding by adult *Heliconius*, a unique trait found in no other Lepidoptera (*Gilbert, 1972*). Our phylogeny indicates that the aoede clade, comprising *H. aoede* and three other species, is likely the earliest-branching lineage of *Heliconius*, followed by the erato-sara clade. This branching order is consistent with a phylogeny based on morphological and behavioral characters (*Penz, 1999*). Since the aoede clade is the only group of *Heliconius* that does not feed on pollen (*Brown, 1981*), the aoede-early scenario (*Figure 1C*) suggests that unique pollen feeding traits arose once after the divergence of the aoede clade (*Figure 4*). In contrast, the erato-early scenario (*Figure 1B*) is less parsimonious as it requires an additional loss of pollen feeding ability in the aoede clade (*Cicconardi et al., 2023*).

*Turner, 1976* named the aoede clade as subgenus *Neruda* based on distinct morphology in all life stages, such as pupae that are more similar to those of *Eueides* (a sister genus of *Heliconius*), unique larva morphology, white eggs like those of *Eueides* (instead of yellow eggs in all other *Heliconius*), a different genus of larval host plant, *Dilkea* (*Benson et al., 1975*) (all other *Heliconius* species feed on *Passiflora*), and chromosome numbers of n = 21–31 which are variable and intermediate between *Heliconius* (n = 21 in most species) and *Eueides* (n = 31) (*Brown et al., 1992*). For these reasons, *Brown, 1981* later raised *Neruda* from subgenus to genus.

Thus, morphological/behavioral/ecological characters tend to support early divergence of the aoede clade (*Brown, 1972*; *Penz, 1999*). Molecular phylogenetic work, on the other hand, seemed to support the erato-early scenario (*Figure 1B*; *Brower and Egan, 1997*; *Beltrán et al., 2007*; *Kozak et al., 2015*; *Kozak et al., 2021*; *Cicconardi et al., 2023*). Consequently, generic status of *Neruda* was revoked (*Kozak et al., 2015*). Our full-likelihood approach, which accounts for ILS as well as introgression, enables inference of a phylogeny that is more parsimonious with morphology and life history. This result parallels analysis of genomic data from African mosquitoes in the *Anopheles gambiae* species complex, in which coalescent-based likelihood analyses support species trees that are more parsimonious for chromosomal rearrangement data (*Thawornwattana et al., 2018*), while prior sliding-window and concatenation analysis favored trees that are less parsimonious (*Fontaine et al., 2015*).

Our likelihood analyses can thus be used to rescue generic or subgeneric status for *Neruda*. Nevertheless, there remains considerable uncertainty near the base of *Heliconius*. Further whole-genome data from *H. aoede* and related species in the aoede, burneyi and doris clades will likely improve resolution.

## Methods

### Whole-genome sequence data and genotyping

We obtained raw sequencing data from previous studies (*Wallbank et al., 2016*; *Edelman et al., 2019*; see *Supplementary file 1a*) and extracted multilocus data as described previously (*Thawornwattana et al., 2022*) as unphased diploid sequences, retaining heterozygous sites. Sequencing reads were mapped to each of two reference genomes (see below) using bwa mem v0.7.15 (*Li, 2013*) with default parameters, and then sorted using sambamba v0.6.8 (*Tarasov et al., 2015*). *RealignerTargetCreator* and *IndelRealigner* modules in GATK v3.8 were used to improve alignment around indels (*McKenna et al., 2010*; *DePristo et al., 2011*). We called genotypes of each individual using *mpileup* and *call* modules in bcftools v1.5 (*Li et al., 2009*) with the multiallelic-caller model (*call -m*) (*Li, 2011*), and set minimum base and mapping quality to 20. We retained high-quality genotype calls using bcftools *filter* using the following filters: (1) a minimum quality score (QUAL) of 20 at both variant and invariant sites; (2) for each genotype, a genotype quality score (GQ) $\geq$ 20 and a read depth (DP) satisfying max(meanDP/2, $d$) $\leq$ DP $\leq$ 2 meanDP, where $d$ = 12 or 20 depending on the dataset, and meanDP is the sample-averaged read depth. This choice of $d$ was used to retain a large number of loci while maintaining low genotype-calling error rate (*Thawornwattana et al., 2022*). For the Z chromosome in females (which are heterogametic in *Heliconius*), we halved the DP threshold ($d$). Finally, we excluded sites within five base pairs (bp) of indels.

### Multilocus datasets

There are three main datasets in this study: (1) 'etales-9spp' contains eight species (*H. melpomene*, *H. besckei*, *H. numata*, *H. burneyi*, *H. doris*, *H. aoede*, *H. erato,* and *H. sara*), one diploid individual per species, representative of all six major clades of *Heliconius* plus one species from a sister genus *Eueides* (*E. tales*). We used two reference genomes (*H. erato demophoon* v1 and *H. melpomene melpomene* Hmel2.5; see http://ensembl.lepbase.org/index.html) for read mapping, and two choices of the minimum read depth cutoffs (12 and 20) to ensure high-quality genotypes, resulting in four datasets in total. All reference genomes were available from http://lepbase.org. (2) 'hmelv25-res' contains eight species (one individual per species) within the melpomene-silvaniform clade (*H. melpomene*, *H. cydno*, *H. timareta*, *H. besckei*, *H. numata*, *H. hecale*, *H. elevatus,* and *H. pardalinus*), mapped to Hmel2.5 reference. The DP threshold ($d$) for genotype calling was 20. (3) To understand the history of the 15b inversion region better, we also compiled a third multilocus dataset for chromosome 15 comprising the pardalinus-hecale clade, *H. numata* with and without the P$_1$ inversion, as well as *H. ismenius* (sister species of *H. numata*) and *H. ethilla* (sister to the pardalinus-hecale clade) (*Supplementary file 1a*, 'silv_chr15' dataset; see below under 'Chromosome 15 inversion region: dataset and analysis') using publicly available data independent of our previous datasets used in this article (*Jay et al., 2018*; *Jay et al., 2021*).

We generated coding and noncoding multilocus datasets from each dataset (*Supplementary file 1a*) as follows (*Appendix 1—figure 3*). First, we extracted coordinates of coding and noncoding loci from the reference genome. In this study, loci refer to short segments of DNA that are far apart. The MSC model implemented in BPP assumes complete linkage within a locus and free recombination between loci. In simulations, species tree inference under MSC is found to be robust to within-locus recombination with recombination rates up to 10× the human rate (*Zhu et al., 2022*). Each coding locus coincided with a protein coding sequence (CDS) and had length at least 100 bp, whereas a noncoding locus can contain noncoding exons, introns, and intergenic regions, and had length 100–1000 bp. Since linkage disequilibrium in *Heliconius* species decays rapidly to background level within 10 kb (*Heliconius Genome Consortium, 2012*), we spaced loci ≥2 kb apart, each assumed approximately independent, to obtain sufficient data. We excluded as loci any repetitive regions annotated in the reference genome. We processed each locus by removing sites containing missing data: the locus was discarded if it consisted of >50% missing data. After filtering, we also discarded loci with 10 or fewer sites remaining. We obtained >10,000 loci in each dataset; see *Supplementary file 1c and I* for the number of loci. For the dataset with read depth cutoff of 12 aligned to the *H. erato* reference, we obtained about 19,000 noncoding loci and 48,000 coding loci. Note that we here obtain more coding than noncoding loci because noncoding loci were more difficult to align with the divergent *Eueides* outgroup (~7% divergent). Filtered noncoding loci were more conserved than coding loci for the same reason. The number of informative sites per locus was 10 for coding loci and 4 for noncoding loci on

average. Average heterozygosity per site was about 0.43% for coding loci and 0.49% for noncoding loci, with *H. besckei* having the lowest heterozygosity (0.15–0.25%) and *H. burneyi* and *E. tales* having the highest heterozygosity (0.7–0.8%).

For the 'etales-9spp' dataset, we separated out inversion regions on chromosomes 2, 6, 13, and 21 into 2b, 6b, 6c, 13b, and 21b (with two adjacent inversions in chromosome 6; chromosome 21 is the Z [sex] chromosome) while flanking regions were denoted 2a, 2c, 6a, 6d, 13a, 13c, 21a, and 21c, resulting in 30 chromosomal regions in total. These inversions were first identified in a previous study (*Seixas et al., 2021*); see coordinates in *Supplementary file 1b*. We obtained >11,000 noncoding loci and >31,000 coding loci in the smallest dataset (aligned to the *H. erato demophoon* reference, *d* = 20) and >31,000 noncoding loci and >48,000 coding loci in the largest dataset (Hmel2.5 reference, *d* = 12); see *Supplementary file 1c*. The median number of sites was 100–130 depending on the dataset. The number of informative sites had median of 2–3 (range: 0–58) per locus for the noncoding loci and 4–5 (0–570) for the coding loci. Again, noncoding loci are underrepresented in our datasets and they tended to be more conserved.

For the 'hmvelv25-res' dataset, we split chromosomes 2 and 15 into inversion (2b and 15b) and flanking regions (2a, 2c, 15a, and 15c), resulting in 25 chromosomal regions in total; coordinates are in *Supplementary file 1b*. Although only the chromosome 15b inversion region has been hitherto identified in this melpomene-silvaniform clade, we wished to test whether the chromosome 2b inversion identified in the erato-sara clade was also present in this group. We obtained >80,500 noncoding loci and >73,200 coding loci for 'hmvelv25-res' (*Supplementary file 1l*). The median number of sites was 339 (range: 11–997) for noncoding loci and 147 (11–12,113) for coding loci. The median number of informative sites per locus was 6 (0–46) for noncoding loci and 2 (0–253) for coding loci.

## Overview of analysis approach

We first used the MSC model without gene flow to explore variation in genealogical history across the genome. We then formulated MSC models with introgression (MSC-I) based on a parsimony argument to explain major patterns of genealogical variation. We estimated the direction, timing, and intensity of introgression under each MSC-I model, and assessed most likely introgression scenarios. For gene flow between closely related species that may be on-going, we also used an MSC model with continuous migration (isolation-with-migration; IM) to estimate rates and directionality of gene flow.

## Species tree estimation under the MSC model without gene flow using BPP

We performed Bayesian inference of species trees under the MSC model without gene flow using BPP v4.4.0 (*Yang and Rannala, 2014*; *Rannala and Yang, 2017*; *Flouri et al., 2018*). This model accounts for gene-tree heterogeneity due to deep coalescence. Hence, the genome-wide variation in estimated genealogy is most likely due to differential gene flow. We grouped loci into blocks of 200 and estimated a posterior distribution of species trees for each block. This blockwise analysis allows us to explore genealogical variation along each chromosomal region and to choose models of introgression for estimation in later analysis. Blocks of coding and noncoding loci were analyzed separately.

The MSC model without gene flow has two types of parameters: species divergence times ($\tau$) and effective population sizes ($\theta = 4N\mu$), both measured in expected number of mutations per site. For the 'etales-9spp' dataset (all four versions; *Figure 1* used *H. erato* reference depth greater than 12, 'minDP12'), we assigned a diffuse gamma prior to the root age $\tau_0 \sim G(7, 200)$, with mean 0.035, and to all population sizes $\theta \sim G(4, 200)$, with mean 0.02. Given $\tau_0$, other divergence times were assigned a uniform-Dirichlet distribution (*Yang and Rannala, 2010*, eq. 2). For each block of loci, we performed 10 independent runs of MCMC, each with $2 \times 10^6$ iterations after a burn-in of $10^5$ iterations, with samples recorded every 200 iterations. We assessed convergence by comparing the posterior distribution of species trees among the independent runs. Nonconvergent runs were discarded. The samples were then combined to produce the posterior summary such as the MAP tree. There were 1355 blocks in total from all four versions of the dataset (*Supplementary file 1c*), so there were 13,550 runs in total. Each run took about 20–30 hr.

Similarly, for the 'hmelv25-res' dataset, we used $\tau_0 \sim G(4, 200)$, with mean 0.02, and population sizes $\theta \sim G(2, 200)$ for all populations, with mean 0.01. Each of the 10 independent runs of the MCMC

took $1 \times 10^6$ iterations after a burn-in of $10^5$ iterations, with samples recorded every 100 iterations. There were 7830 runs in total. Each run took about 15–20 hr.

## Migration rate estimation under the MSC-M model for species triplets using 3s

To obtain more direct evidence of gene flow, we estimated migration rates between all pairs of *Heliconius* species in the 'etales-8spp' dataset under an MSC-with-migration (MSC-M or IM) model using the maximum likelihood program 3s v3.0 (***Dalquen et al., 2017***). The implementation in 3s assumes a species phylogeny of three species $((S_1, S_2), S_3)$ with continuous gene flow between $S_1$ and $S_2$ since their divergence at constant rates in both directions, and requires three-phased haploid sequences per locus. Since our multilocus data were unphased diploid, we phased the data using PHASE v2.1.1 (***Stephens et al., 2001***) to obtain two haploid sequences per individual at each locus. At each locus, we then sampled three types of sequence triplets 123, 113, and 223 with probabilities 0.5, 0.25, and 0.25, respectively, where 113 means two sequences from $S_1$ and one sequence from $S_3$ chosen at random, etc. We used *E. tales* as the outgroup $(S_3)$ for all pairs $(S_1, S_2)$. We analyzed coding and noncoding loci on the autosomes and three regions of the Z chromosome (chromosome 21) separately, each with 28 pairs among the eight *Heliconius* species. Additionally, we analyzed each autosomal region separately. This analysis was done with two choices of reference genome at read depth cutoff $(d)$ of 12.

We fitted two models to each dataset: an MSC without gene flow (M0) and a bidirectional IM (M2). Model M0 has six parameters: two species divergence times ($\tau_1$ for $S_1$–$S_2$ divergence, $\tau_0$ for the root) and four population sizes ($\theta_1$ for $S_1$, $\theta_2$ for $S_2$, $\theta_4$ for the root, and $\theta_5$ for the ancestor of $S_1$ and $S_2$); there is no $\theta_3$ for $S_3$ because there is at most one sequence from $S_3$ per locus. Model M2 has two additional parameters: $M_{12}$ and $M_{21}$, where $M_{12} = m_{12}N_2$ is the expected number of migrants from $S_1$ to $S_2$ per generation, $m_{12}$ is the proportion of migrants from $S_1$ to $S_2$ and $N_2$ is the effective population size of $S_2$. $M_{21}$ is defined similarly. For each model, we performed 10 independent runs of model fitting and chose the run with the highest log-likelihood. We discarded runs with extreme estimates (close to boundaries in the optimization). We then compared models M0 and M2 via likelihood ratio test (LRT) using a chi-squared distribution with two degrees of freedom as a null distribution at a significance threshold of 1%. Adjusting this threshold to account for multiple testing did not change our conclusions because the LRT values were usually extreme, especially in the analysis of all autosomal loci (***Supplementary file 1h***). There were 31 (30 chromosomal regions + all of autosomal loci together) × 2 (coding and noncoding) × 28 (species pairs) × 2 (choices of reference genome) × 10 (replicates) = 34,720 runs in total. For our largest dataset with >46,000 loci, each run of fitting two models (M0 and M2) took 2–3 hr.

We tested between two competing scenarios (***Figure 1B and C***) to infer which was more likely by estimating the internal branch length ($\Delta\tau = \tau_0 - \tau_1$) when either *H. aoede* or *H. erato* was used as an outgroup, with ingroup species representing the melpomene-silvaniform clade. To this end, we compiled another dataset similar to 'hmelv25-res' but included species from all six major clades of *Heliconius* (***Supplementary file 1a***, 'hmelv25-all' dataset). We followed the same procedure as described above. There were 55 pairs in total for each choice of the outgroup species. Estimates of internal branch length close to zero (with the resulting tree becoming star-shaped) suggest that the specified species tree $((S_1, S_2), S_3)$ was likely incorrect. There were 26 (25 chromosomal regions + all autosomal loci together) × 2 (coding and noncoding) × 55 (species pairs) × 2 (choices of reference genome) × 10 (replicates) = 57,200 runs in total.

## Parameter estimation under MSC-I using BPP

Given the species-tree models with introgression of ***Figure 1D and E***, we estimated introgression probabilities ($\varphi$), species divergence times and introgression times ($\tau$), and effective population sizes ($\theta$) for each coding and noncoding dataset from each chromosomal region using BPP v4.6.1 (***Flouri et al., 2020***). We assumed that population sizes of source and target populations remained unchanged before and after each introgression event (thetamodel = linked msci). There were 25 unique parameters in total. We used the same prior distributions for $\tau$ and $\theta$ as in the MSC analysis without gene flow above, with root age $\tau_0 \sim G(7, 200)$ and $\theta \sim G(4, 200)$ for all populations. We assigned a uniform prior $U(0,1)$ to all introgression probabilities ($\varphi$). For each chromosomal region, we performed 10

independent runs of MCMC, each with $1 \times 10^6$ iterations after a burn-in of $10^5$ iterations, with samples recorded every 100 iterations. We assessed convergence by comparing the posterior estimates among the independent runs. Non-convergent runs were discarded. Samples were then combined to produce posterior summaries. Multiple posterior peaks, if they existed, were recorded and processed separately. There were 30 (chromosomal regions) × 2 (coding and noncoding) × 2 (trees 1 and 3) × 10 (replicates) = 1200 runs in total. Each run took 200–300 hrs.

For the two MSC-I models of *Figure 1D and E*, we also estimated the marginal likelihood for each model using thermodynamic integration with 32 Gaussian quadrature points in BPP (*Lartillot and Philippe, 2006*; *Rannala and Yang, 2017*) and calculated Bayes factors to compare the two models (*Supplementary file 1k*). This was done for each coding and noncoding dataset from each chromosomal region. Since the estimates of marginal likelihoods can be noisy, occasionally with extreme outliers, we adjusted estimates of Bayes factor by fitting local quadratic polynomials with span of 0.4 to the difference in the mean log likelihood from the two models, using the loess function in R (*Appendix 1—figure 4*). Additionally, we checked for reliability of the Bayes factor estimates by performing replicate calculation of the Bayes factor for a few chromosomes (3, 4, 9, and 21a) and found the preferred model choice to be reliable. The replicate estimates of log Bayes factor were –8.44, 19.35, –6.65, and 51.45 for chromosomes 3, 4, 9, and 21a, respectively, while the raw estimates were –11.41, 19.63, –10.94, and 53.12 and the adjusted estimates were –2.15, 20.83, –8.84, and 37.80 (*Supplementary file 1k*).

The MSC-I model of *Figure 2B* has six pairs of bidirectional introgression events, with 12 introgression probabilities ($\varphi$), 13 species divergence times and introgression times ($\tau$), and 15 population size parameters ($\theta$), a total of 40 parameters. We assigned the same prior distributions to $\tau_0$ and $\theta$ as in the MSC analysis above, and $\varphi \sim U(0,1)$ for all introgression probabilities. The MSC-I model of *Figure 2C* has three more bidirectional introgression edges, with nine additional parameters (six introgression probabilities and three introgression times). Other settings were the same as above. There were 25 (chromosomal regions) × 2 (coding and noncoding) × 10 (replicates) = 500 runs in total. Most runs took 20–40 d.

## Parameter estimation under an MSC-M model using BPP

In the MSC-I model of *Figure 2B*, each of the pardalinus-hecale and cydno-melpomene clades has very recent estimated introgression times; thus gene flow may be ongoing. We therefore used the MSC-M model to estimate six continuous migration rates between the three species in each clade using BPP v4.6.1 (*Flouri et al., 2023*), assuming the species tree as in *Figure 2*. We assigned prior distributions $\tau_0 \sim G(2, 200)$ for the root age (mean 0.01), $\theta \sim G(2, 500)$ for all populations (mean 0.004), and $M \sim G(2, 10)$ for all migration rates (mean 0.2). The MCMC setup was the same as in the MSC-I analysis above. There were 25 (chromosomal regions) × 2 (coding and noncoding) × 10 (replicates) × 2 (pardalinus-hecale and cydno-melpomene clades) = 1000 runs in total. Each run took 120–150 hr.

## Chromosome 15 inversion region: Dataset and analysis

To investigate the evolutionary history of the inversion region in chromosome 15 (15b region or *P* locus), we obtained genomic sequence data for six species (*H. numata* homozygous for the ancestral orientation, *H. numata* homozygous for the inversion, *H. ismenius*, *H. pardalinus*, *H. elevatus*, *H. hecale*, and *H. ethilla*), with two individuals per species, from previous studies (*Jay et al., 2018*; *Jay et al., 2021*; *Supplementary file 1a*, 'silv_chr15' dataset). We extracted a multilocus dataset using an improved pipeline compared with that described above (see 'Whole-genome sequence data and genotyping' and 'Multilocus datasets'). Here, a different filtering strategy used more complex, multi-stage genotyping to account for multiple individuals per species as follows. First, we removed Illumina adapters and trimmed low quality bases using trimmomatic v.0.39 (SLIDINGWINDOW:4:20 MINLEN:50). Next, sequencing reads were mapped to the *H. melpomene melpomene* reference assembly (Hmel2.5) using bwa-mem v.0.7.17 (*Li, 2013*). Duplicate reads were masked using MarkDuplicates (Picard) in GATK v.4.2.6.1 (*Poplin et al., 2018*). We jointly called genotypes on chromosome 15 of individuals from the same species (and the same inversion genotype) using *mpileup* and *call* modules in bcftools v1.17 (*Li et al., 2009*) with the multiallelic-caller model (*call -m*) (*Li, 2011*), and set the minimum base and mapping quality to 20. Only high-quality SNPs (QD score ≥2.0 and MQ score ≥40) were retained.

To obtain multilocus data, genomic coordinates of coding and noncoding loci were obtained from the reference genome as described above. Noncoding loci were 100–1000 bp in length and at least 2 kb apart. Coding loci were at least 100 bp in length (no maximum length limit) and at least 2 kb apart. To maximize information, minimum spacing was enforced for loci within the inversion region. We then extracted sequence alignments for each locus using the following procedure. All SNPs passing the quality filter were included. Constant sites were obtained from the reference sequence unless they were masked by one of the following criteria: (1) read depth below 20 (coded as '–') and (2) non-SNP variant or low-quality SNP (coded as 'N'). For each locus, we excluded sequences with >50% of sites missing ('–' or 'N'), and excluded sites with all missing data. We discarded loci with only a single sequence remaining after filtering. Finally, we grouped loci into three regions as before: one inversion region (15b) and two flanking regions (15a and 15c). We obtained 218, 95, and 960 coding loci and 424, 368, and 2446 noncoding loci in 15a, 15b, and 15c regions, respectively.

We performed blockwise estimation of species trees under the MSC model without gene flow as described earlier in blocks of 100 and 200 loci (see 'Species tree estimation under the MSC model without gene flow using BPP'). For the 200-locus blocks, there were 23 (7 coding blocks + 16 noncoding blocks) × 10 (replicates) = 230 runs; each run took about 60 hr. For the 100-locus blocks, there were 45 (13 coding blocks + 32 noncoding blocks) × 10 (replicates) = 450 runs; each run took about 30 hrs.

## Acknowledgements

This study was supported by Harvard University (YT, FS, and JM), and by Biotechnology and Biological Sciences Research Council grants (BB/T003502/1, BB/X007553/1 and BB/R01356X/1) and Natural Environment Research Council grant (NSFDEB-NERC NE/X002071/1) to ZY

## Additional information

### Funding

| Funder | Grant reference number | Author |
| --- | --- | --- |
| Biotechnology and Biological Sciences Research Council | BB/T003502/1 | Ziheng Yang |
| Biotechnology and Biological Sciences Research Council | BB/X007553/1 | Ziheng Yang |
| Biotechnology and Biological Sciences Research Council | BB/R01356X/1 | Ziheng Yang |
| Natural Environment Research Council | NE/X002071/1 | Ziheng Yang |

The funders had no role in study design, data collection and interpretation, or the decision to submit the work for publication.

### Author contributions

Yuttapong Thawornwattana, Formal analysis, Investigation, Visualization, Methodology, Writing - original draft, Writing – review and editing; Fernando Seixas, Data curation, Writing – review and editing; Ziheng Yang, Conceptualization, Software, Supervision, Funding acquisition, Investigation, Methodology, Writing – review and editing; James Mallet, Conceptualization, Supervision, Funding acquisition, Investigation, Methodology, Writing – review and editing

### Author ORCIDs

Yuttapong Thawornwattana http://orcid.org/0000-0003-2745-163X
James Mallet https://orcid.org/0000-0002-3370-0367

Reviewer #1 (Public Review): https://doi.org/10.7554/eLife.90656.3.sa1

Reviewer #2 (Public Review): https://doi.org/10.7554/eLife.90656.3.sa2
Reviewer #3 (Public Review): https://doi.org/10.7554/eLife.90656.3.sa3
Author Response https://doi.org/10.7554/eLife.90656.3.sa4

## Additional files

### Supplementary files

• Supplementary file 1. Supplementary tables. (**a**) Sample information. (**b**) Coordinates of chromosome inversion regions considered in this study. (**c**) Number of coding and noncoding loci by chromosomal region in four versions of the 'etales-9spp' dataset. (**d**) Proportions of posterior species tree estimates (maximum a posteriori [MAP] trees) from BPP multispecies coalescent (MSC) analysis without gene flow of all four versions of the 'etales-9spp' dataset in blocks of 200 loci, with BNM clade (*H. besckei*, *H. numata*, *H. melpomene*) collapsed into a single tip, summarized into major regions: auto (all autosomes excluding inversion regions), the Z chromosome (chr 21, excluding the inversion region), and individual inversion regions (2b, 6b, 6c, 13b, and 21b). n = number of blocks. Tree indices correspond to those in *Figure 1—figure supplement 2*. A full list of MAP trees for each chromosomal region is provided in (e). Full results without BNM lumping are provided in (f and g). (**e**) Proportions of MAP trees from BPP MSC analysis without gene flow of all four versions of the 'etales-9spp' dataset in blocks of 200 loci, with BNM clade collapsed into a single tip, summarized by chromosomal region. Genome-wide summaries are given in (d). (**f**) Full results from BPP MSC analysis without gene flow of the 'etales-9spp' dataset, summarized into major regions of the genome. See also legend to (d). Trees are in a decreasing order of the total frequencies across all four versions of the dataset. Tree indices correspond to those in *Figure 1—figure supplement 1*. (**g**) Full results from BPP MSC analysis without gene flow of the 'etales-9spp' dataset, summarized by chromosomal region. See legend to (f). (**h**) Bayes factor $B = M_{1E}/M_{1D}$ for comparing the two MSC-I models in *Figure 1D* ($M_{1E}$; scenario 1) and *Figure 1E* ($M_{1D}$; scenario 2), calculated for coding and noncoding loci for each chromosomal region using thermodynamic integration with 32 Gaussian quadrature points. $M_{1E}$ (scenario 2) is preferred if $B>100$, $M_{1D}$ (scenario 1) is preferred if $B < 0.01$, and the test is not significant (n.s.) otherwise (i.e. $100 < B < 0.01$). The last three columns are adjusted values of Bayes factor to account for numerical instability of the thermodynamic integration estimates (see 'Methods' and *Appendix 1—figure 4*). (**i**) Posterior means and 95% HPD intervals (in parentheses) of parameters from the BPP MSC-I model in *Figure 1D* (scenario 1) using noncoding and coding loci for each chromosomal region of 'etales-8spp' dataset (see a). Multiple posterior peaks, if present, are reported, for example there are two peaks, denoted chr1 and chr1-alt, from the coding loci in chromosome 1. Plots of these estimates with 95% HPD intervals are provided in *Figure 1—figure supplement 3*. Plots of species tree with estimated divergence times, introgression times, and introgression probabilities are provided in *Figure 1—figure supplement 5*. (**j**) Posterior means and 95% HPD intervals (in parentheses) of parameters from the BPP MSC-I model in *Figure 1E* (scenario 2) using noncoding and coding loci for each chromosomal region. Multiple posterior peaks, if present, are reported, for example, there are two peaks, denoted chr1 and chr1-alt, from the coding loci in chromosome 1. Plots of these estimates with 95% HPD intervals are provided in *Figure 1—figure supplement 3*. Plots of the species tree with estimated divergence times, introgression times, and introgression probabilities are in *Figure 1—figure supplement 6*. (**k**) Maximum likelihood estimates (MLEs) of parameters under the isolation-with-migration (IM) model obtained from 3s for each of the 28 pairs of species in the 'etales-9spp' dataset (*H. erato* reference, minDP12), using *Eueides tales* (Tal) as an outgroup. We used the likelihood ratio test (LRT) statistic to test if the model with gene flow (M2) was preferred to the model without gene flow (M0) using the p-value threshold of 1%. The 'auto' dataset used all autosomal loci excluding the inversion regions (see c for the number of loci). Plots of migration rate and divergence time estimates are provided in *Appendix 1—figure 1*. Plots of all parameter estimates with confidence intervals are provided in *Figure 1—figure supplement 7*. (**l**) Number of coding and noncoding loci in each chromosomal region in the 'hmelv25-res' dataset used in BPP analysis and the 'hmelv25-all' dataset use in 3s analysis. (**m**) Proportions of MAP trees, with minimum, median, and maximum posterior probabilities shown in parentheses, from BPP MSC analysis without gene flow of the 'hmelv25-res' dataset in blocks of 200 loci, summarized into four major regions: auto (all autosomes excluding 2b and 15b), 2b and 15b inversion regions, and chromosome 21 (Z chromosome). Trees are in a decreasing order of combined frequency across all chromosomes. Tree indices correspond to those in *Figure 2—figure supplement 1*. (**n**) Proportions of MAP trees, with minimum, median, and maximum posterior probabilities shown in parentheses

from BPP MSC analysis without gene flow in blocks of 200 loci, summarized by chromosomal region. See legend to (m). (**o**) MLEs of parameters under the MSC-M model obtained from 3s for each of the 55 pairs of species in the 'hmelv25-all' dataset using *H. aoede* as an outgroup. We used the LRT statistic to test if the model with gene flow (M2) was preferred to the model without gene flow (M0) using the p-value threshold of 1%. See *Figure 2—figure supplement 2A* for plots of these estimates. The 'auto' setting used all autosomal loci excluding the inversion regions. See l for the number of loci. (**p**) MLEs of parameters under the MSC-M model obtained from 3s for each of the 55 pairs of species in the 'hmelv25-all' dataset using *H. erato* as an outgroup. See legend to (o). See *Figure 2—figure supplement 2* for plots of these estimates. (**q**) Posterior means and 95% HPD intervals (in parentheses) of parameters obtained from the BPP MSC-I model of *Figure 2B* using noncoding loci from each non-15b chromosomal region, and under the MSC-I model m1 of *Figure 3—figure supplement 3* for the chromosome 15b inversion region. Plots of the estimates are provided in *Figure 2—figure supplement 4*. 'n/a' means the parameter does not exist in the model. (**r**) Posterior means and 95% HPD intervals (in parentheses) of parameters from BPP MSC-I models are provided in *Figure 2B* and *Figure 3—figure supplement 3* (model m1) using coding loci. See legend to (q). (**s**) Posterior means and 95% HPD intervals (in parentheses) of parameters from BPP MSC-M models for the pardalinus-hecale clade: ((Par, Ele), Hec), with six continuous migration rates ($M = Nm$) among the three species, two species divergence times ($\tau$) and five population sizes ($\theta$). Plot of the estimates is provided in *Figure 2—figure supplement 6A*. (**t**) Posterior means and 95% HPD intervals (in parentheses) of parameters obtained from BPP MSC-M models for the pardalinus-elevatus clade: (Par, Ele), with two continuous migration rates ($M = Nm$). All populations were assumed to have the same population size ($\theta$). Plot of the estimates is provided in *Figure 2—figure supplement 6B*. (**u**) Posterior means and 95% HPD intervals (in parentheses) of parameters from BPP MSC-M models for the cydno-melpomene clade: ((Tim, Cyd), Mel), with six continuous migration rates ($M = Nm$) among the three species, two species divergence times ($\tau$), and five population sizes ($\theta$). Plot of the estimates is provided in *Figure 2—figure supplement 6C*. (**v**) Proportions of MAP trees, with minimum, median, and maximum posterior probabilities shown in parentheses, from BPP MSC analysis without gene flow of the 'silv-chr15' dataset in blocks of 200 loci or 100 loci, summarized by three regions: the inversion region (15b) and flanking regions (15a and 15c). Trees are in decreasing order of combined frequency across all chromosomal regions and from both options of block size. (**w**) Posterior means and 95% HPD intervals (in parentheses) of parameters obtained from five BPP MSC-I models in *Figure 3—figure supplement 3*.

- MDAR checklist

## Data availability

All multilocus alignment datasets generated in this study (see *Supplementary file 1a, c and I*) and scripts for reproducing main analyses are available in Zenodo at https://doi.org/10.5281/zenodo.8415106.

The following dataset was generated:

| Author(s) | Year | Dataset title | Dataset URL | Database and Identifier |
|---|---|---|---|---|
| Thawornwattana Y, Seixas F, Yang Z, Mallet J | 2023 | Major patterns in the introgression history of Heliconius butterflies | https://zenodo.org/records/8415106 | Zenodo, 10.5281/zenodo.8415106 |

The following previously published datasets were used:

| Author(s) | Year | Dataset title | Dataset URL | Database and Identifier |
|---|---|---|---|---|
| Edelman N | 2019 | Genomic architecture and introgression shape a butterfly radiation | https://www.ncbi.nlm.nih.gov/bioproject/?term=PRJNA532398 | NCBI BioProject, PRJNA532398 |
| Jay P | 2018 | Whole genome sequence data from silvaniform Heliconius individuals | https://www.ncbi.nlm.nih.gov/bioproject/?term=PRJNA471310 | NCBI BioProject, PRJNA471310 |
| Jay P | 2021 | Heliconius numata individual whole genome shotgun sequencing dataset | https://www.ncbi.nlm.nih.gov/bioproject/?term=PRJEB40136 | NCBI BioProject, PRJEB40136 |

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

# Appendix 1

## Major introgression patterns in the melpomene-silvaniform clade inferred using 3s and BPP

To assess the possibility of introgression that might support the autosome-majority tree or the Z chromosome tree, we estimate pairwise gene flow rates under the MSC-M model between each pair of species in the melpomene clade as well as *H. burneyi*, *H. doris,* and *H. erato* using the program 3s (*Zhu and Yang, 2012*; *Dalquen et al., 2017*). This implementation allows for continuous bidirectional gene flow between two species since divergence. We performed maximum likelihood estimation and likelihood ratio tests (LRT) to determine if a allowing for bidirectional gene flow (model M2) is a better fit than without gene flow (model M0). An outgroup species was used to improve the power of the test but it was not involved in gene flow. We used *H. aoede* as an outgroup in all species pairs. We find gene flow to be prevalent in the autosomes but almost absent in the Z chromosome (*Appendix 1—figure 2A*). In particular, we find no strong evidence of gene flow in the Z chromosome between *H. numata* and *H. besckei*, which is required to explain the Z chromosome tree if the autosome-majority tree is the true species tree. Taken together, these results suggest that the Z chromosome tree represents the true species branching order, and autosomal trees are a result of extensive introgression throughout the history of this group. Using *H. erato* as an outgroup instead of *H. aoede* leads to the same conclusion (*Appendix 1—figure 2*), although the internal branch length becomes zero whenever one of the ingroup species is *H. aoede*. A full list of estimates from all species pairs is provided in *Figure 2—figure supplement 2* and *Supplementary files 1o and p*.

Next, we formulated two models to capture major introgression events in the group using the Z chromosome tree as a base tree (*Figure 2B and C*). The first model is the main model used for most parts of the genome and had six pairs of bidirectional introgression (*Figure 2B*): two for explaining different positions of *H. numata* (nodes n3–tcmeph1 and n2–eph1, relating trees 1–3, tree 4 and trees 5–7), one for *H. pardalinus–H. elevatus* gene flow, which is ongoing in sympatric populations (p1–e1) (*Rosser et al., 2019*), two for gene flow involving *H. melpomene* with *H. cydno* (m1–c1) and with *H. timareta* (m2–t1) (explaining variation among trees 1–3 and trees 5–7), and one for the possibility of *H. besckei* and *H. numata* (b1–n1) as suggested by the relationship between the Z chromosome tree and autosome trees. To keep the model tractable, we excluded potential *H. cydno–H. timareta* introgression in the model because these species are presently allopatric and signals of gene flow between them are likely to be via *H. melpomene*. We also assumed that population size did not change after introgression. This model has 12 introgression probabilities ($\varphi$), 13 species divergence times and introgression times ($\tau$), and 15 population size parameters ($\theta$), a total of 40 parameters. The second model is for the chromosome 15 inversion region (15b), which had three additional pairs of introgression edges, with 9 additional parameters (*Figure 3—figure supplement 3*, model m1). We estimated those parameters for coding and noncoding loci of each chromosomal region using BPP (*Flouri et al., 2020*).

Our estimated introgression histories are largely consistent across the genome and between coding and noncoding regions (*Figure 2—figure supplements 3 and 4*, *Supplementary file 1q and r*). This pattern agrees with blockwise variation in species relationships (estimated without gene flow, *Figure 2A*). Gene flow was almost absent in the Z chromosome and consistent with our estimates under the MSC-M model.

The two introgression events involving *H. numata* that resolve majority autosomal relationships (trees 1–3, 5–7) and the Z chromosome tree (tree 4) were estimated to have high probabilities except for on the Z chromosome and in the chromosome 15b inversion region (*Figure 2—figure supplement 4A*). First, for the oldest introgression between *H. numata* and the common ancestor of the cydno-melpomene + pardalinus-hecale clade, the introgression is estimated to be unidirectional into *H. numata*, with probability ($\varphi_{n3 \leftarrow tcmeph1}$) ~ 0.85 in most parts of the genome while the introgression in the opposite direction was almost absent ($\varphi_{tcmeph1 \leftarrow n3} < 0.05$). Second, the subsequent introgression between *H. numata* and the common ancestor of the pardalinus-hecale clade is estimated to be bidirectional in many chromosomes, with probabilities ($\varphi_{n2 \leftarrow eph1}$ and $\varphi_{eph1 \leftarrow n2}$) ~ 0.3 on average in both directions, although values of these probabilities varied considerably among chromosomes (*Figure 2—figure supplement 4A*).

Within the pardalinus-hecale clade, we estimate introgression probabilities between the sister species *H. pardalinus* and *H. elevatus* ($\varphi_{e1 \leftarrow p1}$ and $\varphi_{p1 \leftarrow e1}$) to be ~0.5 in both directions in most

chromosomes, although there is some variation across the genome. The time of this introgression ($\tau_{e1}$ or $\tau_{p1}$) is essentially zero (*Figure 2—figure supplement 4A*, *Supplementary file 1q and r*), consistent with the fact that these two species still hybridize today (*Mallet et al., 2007*). In some chromosomes such as 13 and 21, posterior estimates of ~0.5 had wide intervals (*Figure 2—figure supplement 4A*). The uncertainty may be due to weak evidence for gene flow, even with non-zero posterior means.

Within the cydno-melpomene clade, both *H. melpomene–H. cydno* and *H. melpomene–H. timareta* introgression is estimated to be highly asymmetrical, with *H. melpomene* being the main recipient in both cases, and with the introgression probabilities varying widely across the genome (*Figure 2—figure supplement 4A*). The *H. cydno → H. melpomene* introgression is estimated to be strictly unidirectional, with introgression probability ($\varphi_{m1\leftarrow c1}$) of ~0.35 in noncoding regions, ~0.25 in coding regions, and zero in the opposite direction ($\varphi_{c1\leftarrow m1}$). For *H. melpomene–H. timareta*, the introgression probability for *H. timareta → H. melpomene* ($\varphi_{m2\leftarrow t1}$) is highly variable across the genome, with values ranging from close to 0 to almost 1; the opposite direction ($\varphi_{t1\leftarrow m2}$) had low probabilities (<0.1) for most chromosomes.

Surprisingly, we found strictly unidirectional introgression from *H. numata* into *H. besckei* in all regions of the genome, including the Z chromosome, with probability ($\varphi_{b1\leftarrow n1}$) around 0.3–0.4 in the noncoding regions and 0.5–0.6 in the coding regions. This is the only substantial introgression estimated in the Z chromosome, with $\varphi_{b1\leftarrow n1} \approx 0.3$, which was still lower than most autosomes. The chromosome 15 inversion region (15b) has a unique history that is much more complex than the rest of the genome; results for 15b are discussed in the next section.

Species divergence time estimates are precise and highly similar across the genome (*Figure 2—figure supplement 4A*). The posterior means from coding and noncoding loci are strongly correlated, with $\tau_{coding} \approx x\ \tau_{noncoding}$ where $x$ varies between 0.5 and 0.6 ($r^2 > 0.95$) in most chromosomal regions (*Figure 2—figure supplement 5*). We estimate the age of the Melpomene clade ($\tau_r$) to be ~0.020 substitutions per site based on noncoding data and ~0.013 based on coding data. Assuming a constant neutral mutation rate of $2.9 \times 10^{-9}$ per site per generation (95% CI: $1.3 \times 10^{-9}$, $1.5 \times 10^{-9}$) and four generations per year (*Keightley et al., 2015*), we obtain a conservative estimate that the melpomene clade diverged about 1.7 (CI: 0.9, 3.8) Ma, which is comparable to a previous estimate of 3.7 (CI: 3.2, 4.3) Ma based on molecular clock dating (*Kozak et al., 2015*). We estimate present-day and ancestral population sizes ($\theta$) to be of the order of 0.01, largely consistent among chromosomes (*Figure 2—figure supplement 4B*, *Supplementary file 1q and r*). For inbred reference individuals (*H. melpomene, H. timareta, H. numata,* and *H. pardalinus*; see *Supplementary file 1a*), the estimates ($\theta_{Mel}, \theta_{Tim}, \theta_{Num,}$ and $\theta_{Par}$) tended to be much more variable, with some chromosomes having unusually low estimates in the order of 0.001 or lower. *H. melpomene* has the lowest population size estimate of about 0.002–0.004 on average. This pattern corresponds well to the fluctuating levels of heterozygosity among chromosomes as a result of inbreeding. Adding more individuals should help stabilize the estimates. Inbred individuals should be avoided in population genomic analyses.

## MSC-M model for pardalinus-hecale and cydno-melpomene clades

In the pardalinus-hecale clade, we find the migration rates ($M_{Par\to Ele}$ and $M_{Ele\to Par}$) between *H. pardalinus* and *H. elevatus* to be high and largely asymmetrical, with $M \approx$ 2–4 migrants per generation in one direction and a lower rate ($M < 1$) in the opposite direction (*Figure 2—figure supplement 6A*), consistent with the introgression probability estimates (*Figure 2—figure supplement 4A*). However, we observe a label-switching problem in which different runs of model fitting on the same data converged to different posterior peaks that differed in the main direction of gene flow accompanied by a flip in the corresponding population size parameters. This near unidentifiability of $M$ and $\theta$ appears to be partly associated with high migration rates and low genomic distinctiveness between *H. pardalinus* and *H. elevatus*. Since these two species appear to be virtually panmictic and to share effective population size over most of their genome, we also fitted an alternative model to these two species assuming all three populations (two present-day and one ancestral) share the same population size. Under this simpler model, the migration rates are still high ($M \sim 1$) but less extreme, and the unidentifiability issue largely disappears (*Figure 2—figure supplement 6B*, *Supplementary file 1t*). We also detect significant but lesser gene flow involving *H. hecale*, with migration rates $M_{Hec\to Ele}, M_{Ele\to Hec}, M_{Hec\to Par},$ and $M_{Par\to Hec}$ <0.1 in most chromosomes (*Figure 2—figure supplement 6A*). Divergence time estimates are more stable among chromosomes and are not affected by the unidentifiability issue. Under this MSC-M model, the divergence time between *H. pardalinus* and

*H. elevatus* is $\tau_{ep} \approx 0.010$ substitutions per site, and the divergence time between *H. hecale* and *H. pardalinus* + *H. elevatus* is $\tau_{eph} \approx 0.013$ substitutions per site based on noncoding data; coding data led to about half these values (*Figure 2—figure supplement 6A*, *Supplementary file 1s*). These estimates are slightly older than those obtained under a larger model assuming pulse introgression in *Figure 2B* are $\tau_{ep} \approx 0.0070$ and $\tau_{eph} \approx 0.0076$ using noncoding data ($\tau_{ep} \approx 0.002$ and $\tau_{eph} \approx 0.003$ using coding data) (*Supplementary file 1q and r*).

In the cydno-melpomene clade, estimated migration rates under the MSC-M model are below 0.1, generally much lower than those in the pardalinus-hecale clade, with large variation among chromosomes (*Figure 2—figure supplement 6C*, *Supplementary file 1u*). The label-switching problem is rare. Chromosomes with unusually low incoming migration rate estimates tend to be associated with low heterozygosity in inbred individuals (*H. melpomene* and *H. timareta*). For instance, the *H. melpomene* individual had almost no variation in chromosomes 2, 9, 10, and 18, and as a result, the migration rates $M_{Cyd \to Mel}$ and $M_{Tim \to Mel}$ (and $\theta_{Mel}$, $\theta_{Tim}$) are estimated about one or two orders of magnitude lower on those chromosomes compared with other chromosomes (*Supplementary file 1u*). Unlike the introgression model where introgression is largely from *H. cydno* and *H. timareta* into *H. melpomene*, there is no clear pattern of highly asymmetrical gene flow into *H. melpomene*. In this MSC-M model, we also allow for gene flow between *H. timareta* and *H. cydno*, which is estimated to be $M_{Cyd \to Tim} \approx 0.1$ and $M_{Tim \to Cyd} \approx 0.04$ migrants per generation in both coding and noncoding loci (*Supplementary file 1u*). Divergence time estimates are largely comparable across chromosomes. However, the estimates tend to be older than those obtained under the introgression model of *Figure 2—figure supplement 4A*. We obtained $\tau_{tc} \approx 0.007$ and $\tau_{tcm} \approx 0.012$–0.014 using coding or noncoding loci under the MSC-M model (*Supplementary file 1u*) while $\tau_{tc} \approx 0.0025$ and $\tau_{tcm} \approx 0.007$ using noncoding data and $\tau_{tc} \approx 0.001$ and $\tau_{tcm} \approx 0.0043$ using coding data (*Supplementary file 1q and r*).

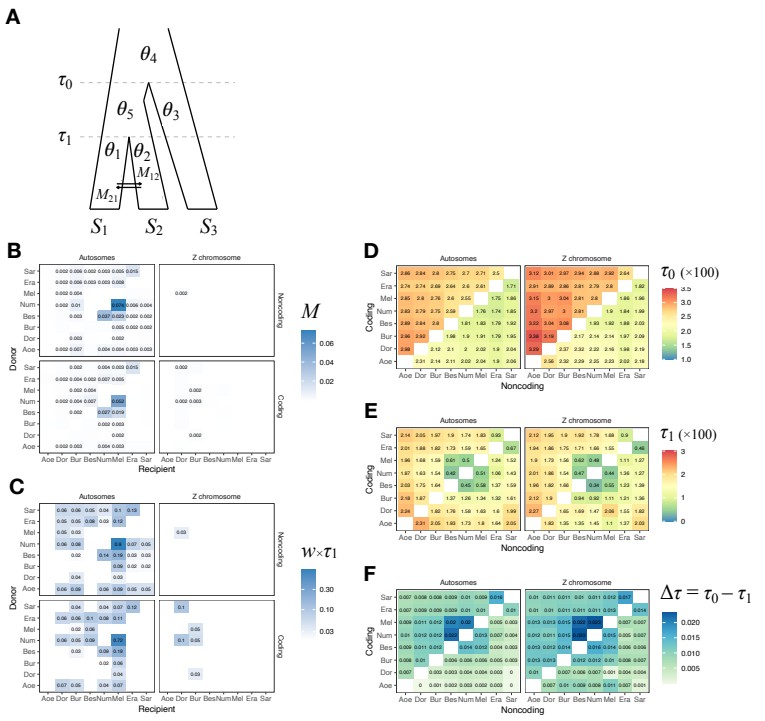

fig S3: 3s, etales-9spp, E. tales outgroup, heradem, minDP12

**Appendix 1—figure 1.** Isolation-with-migration (IM) analysis of the 'etales-9spp' dataset using 3s. (**A**) Model specification in 3s. There are three types of parameters: divergence times ($\tau_1$, $\tau_0$), effective population sizes ($\theta_1$, $\theta_2$, $\theta_4$, and $\theta_5$) and migration rates ($M_{12}$, $M_{21}$). (**B**) Maximum-likelihood estimates (MLEs) of the pairwise migration rates ($M_{12}$, $M_{21}$), with donor species on the y-axis and recipient species on the x-axis. We used *Eueides tales* as
*Appendix 1—figure 1 continued on next page*

*Appendix 1—figure 1 continued*

the outgroup ($S_3$ in **A**). Top and bottom quadrants are results from noncoding and coding loci, respectively. Left quadrants show results from all autosomal loci. Right quadrants show results from all Z chromosome loci (21a + 21b + 21c). For the number of loci, see **Supplementary file 1c** (*H. erato* reference, minDP (d) = 12). For pairs without displayed numbers, the likelihood ratio test (LRT) was not significant at 0.1%, thereby failing to reject a null model of zero gene flow. (**C**) Mutation-scaled migration rates ($w_{12} = 4M_{12}/\theta_2$ and $w_{21} = 4M_{21}/\theta_1$) as a measure of expected admixture fraction in the recipient genome. (**D**) Root age ($\tau_0$). Estimates from coding loci are shown in the upper triangle while estimates from noncoding loci are in the lower triangle. (**E**) Divergence time of the ingroup species pair ($\tau_1$). (**F**) Internal branch length ($\Delta\tau = \tau_0 - \tau_1$). Full results are in **Supplementary file 1k** and **Figure 1—figure supplement 7**.

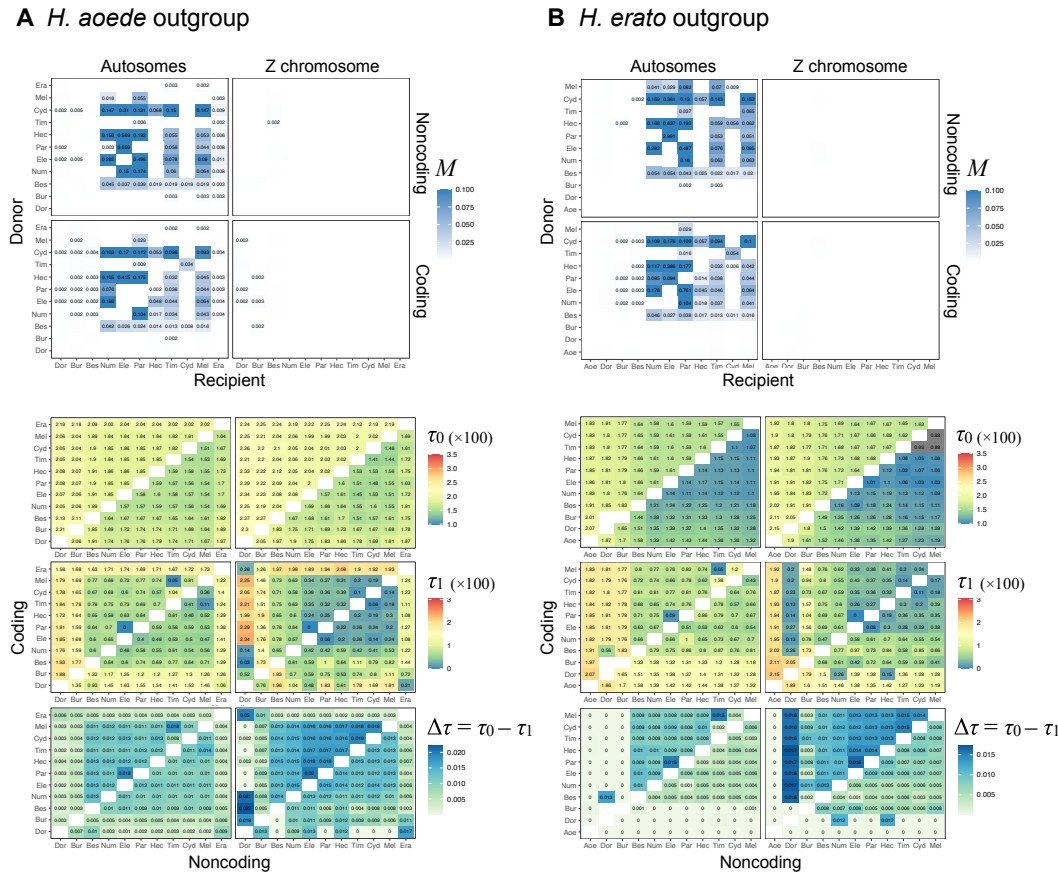

**Appendix 1—figure 2.** Maximum likelihood estimates (MLEs) of migration rates (*M*), divergence times ($\tau_0$, $\tau_1$), and the internal branch length ($\Delta\tau = \tau_0 - \tau_1$) from 3s analysis of the 'hmelv25-all' dataset, using (**A**) *H. aoede* or (**B**) *H. erato* as an outgroup. See legend to **Appendix 1—figure 1**.

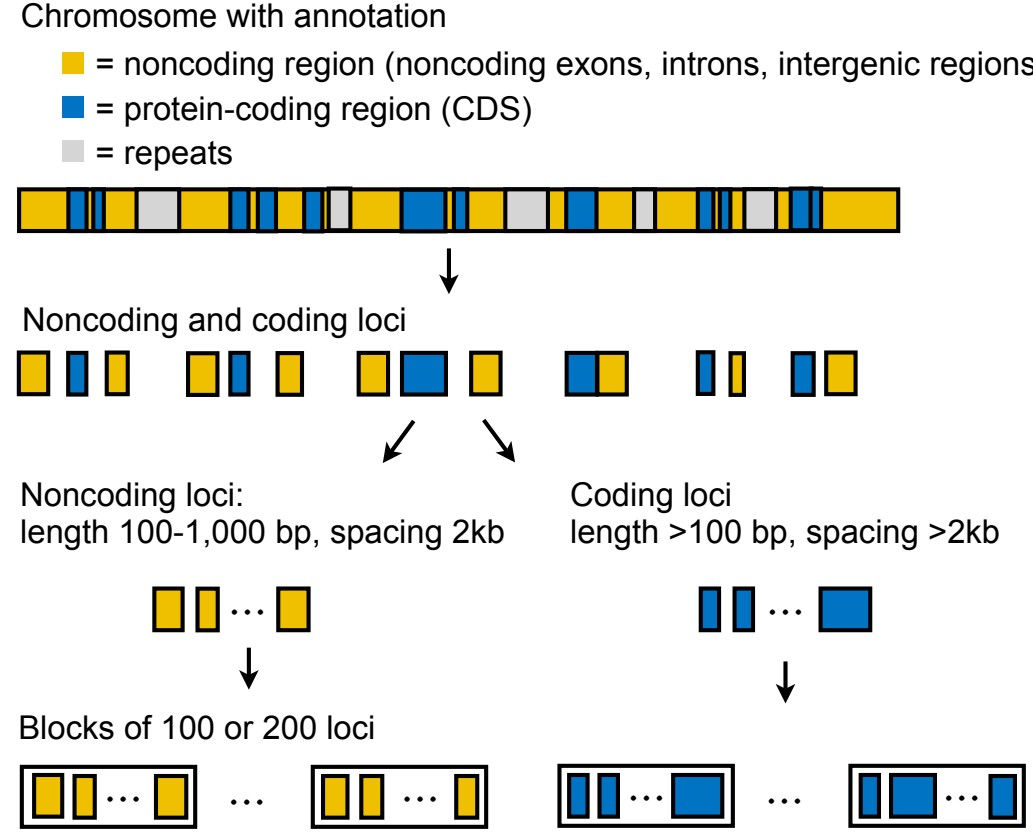

**Appendix 1—figure 3.** Generation of multilocus data. Coordinates of coding and noncoding loci were obtained from a reference genome with an annotation of coding sequences (CDS).

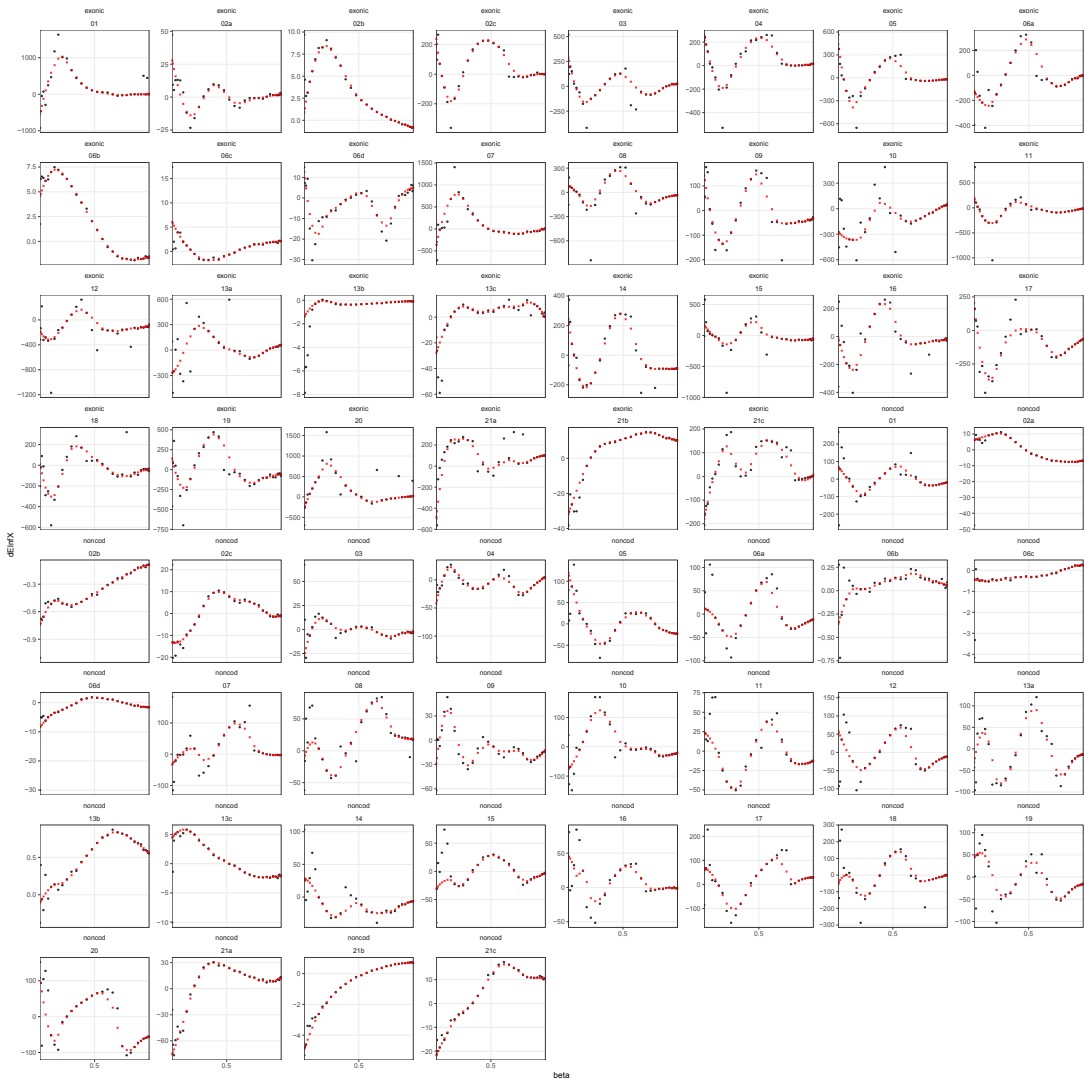

**Appendix 1—figure 4.** Raw (black) and adjusted (red) estimates of the difference in the mean log likelihood from the two models (y-axis). Raw and adjusted estimates of the Bayes factor are in *Supplementary file 1h*.

