## [Editor Report · eLife assessment]

This **important** study revises the evolutionary history of *Heliconius* butterflies, a well-established model system for understanding speciation in the presence of gene flow between species. Using a **convincing** statistical phylogenetic approach that relies on the multispecies coalescent, the authors reconstruct the evolution of the lineage, including the timing of speciation events and the history of gene flow. The new phylogeny will be of interest to all researchers working on *Heliconius* butterflies, and the phylogenetic approach to investigators aiming to understand the history of lineages that have experienced extensive gene flow.

---

## [Referee Report · Reviewer #1 (Public Review)]

Summary:

This study aims to further resolve the history of speciation and introgression in Heliconius butterflies. The authors break the data into various partitions and test evolutionary hypotheses using the Bayesian software BPP, which is based on the multispecies coalescent model with introgression. By synthesizing these various analyses, the study pieces together an updated history of Heliconius, including a multitude of introgression events and the sharing of chromosomal inversions.

Strengths:

Full-likelihood methods for estimating introgression can be very computationally expensive, making them challenging to apply to datasets containing many species. This study provides a great example of how to apply these approaches by breaking the data down into a series of smaller inference problems and then piecing the results together. On the empirical side, it further resolves the history of a genus with a famously complex history of speciation and introgression, continuing its role as a great model system for studying the evolutionary consequences of introgression. This is highlighted by a nice Discussion section on the implications of the paper's findings for the evolution of pollen feeding.

Weaknesses:

The analyses in this study make use of a single method, BPP. The analyses are quite thorough so this is okay in my view from a methodological standpoint, but given this singularity, more attention should be paid to the weaknesses of this particular approach. Additionally, little attention is paid to comparable methods such as PhyloNet and their strengths and weaknesses in the Introduction or Discussion. BPP reduces computational burden by fixing certain aspects of the parameter space, such as the species tree topology or set of proposed introgression events. While this approach is statistically powerful, it requires users to make informed choices about which models to test, and these choices can have downstream consequences for subsequent analyses. It also might not be as applicable to systems outside of Heliconius where less previous information is available about the history of speciation and introgression. In general, it is likely that most modelling decisions made in the study are justified, but more attention should be paid to how these decisions are made and what the consequences of them could be, including alternative models.

• Co-estimating histories of speciation and introgression remains computationally challenging. To circumvent this in the study, the authors first estimate the history of speciation assuming no gene flow in BPP. While this approach should be robust to incomplete lineage sorting and gene tree estimation, it is still vulnerable to gene flow. This could result in a circular problem where gene flow causes the wrong species tree to be estimated, causing the true species tree to be estimated as a gene flow event. This is a flaw that this approach shares with summary-statistic approaches like the D-statistic, which also require an a-priori species tree. Enrichment of particular topologies on the Z chromosome helps resolve the true history in this particular case, but not all datasets will have sex chromosomes or chromosome-level assemblies to test against.

• The a-priori specification of network models necessarily means that potentially better-fitting models to the data don't get explored. Models containing introgression events are proposed here based on parsimony to explain patterns in gene tree frequencies. This is a reasonable and common assumption, but parsimony is not always the best explanation for a dataset, as we often see with phylogenetic inference. In general, there are no rigorous approaches to estimating the best-fitting number of introgression events in a dataset. Likewise, the study estimates both pulse and continuous introgression models for certain partitions, though there is no rigorous way to assess which of these describes the data better.

• Some aspects of the analyses involving inversions warrant additional consideration. Fewer loci were able to be identified in inverted regions, and such regions also often have reduced rates of recombination. I wonder if this might make inferences of the history of inverted regions vulnerable to the effects of incomplete lineage sorting, even when fitting the MSC model, due to a small # of truly genealogically independent loci. Additionally, there are several models where introgression events are proposed to explain the loss of segregating inversions in certain species. It is not clear why these scenarios should be proposed over those in which the inversion is lost simply due to drift or selection.

---

## [Referee Report · Reviewer #2 (Public Review)]

Thawornwattana et al. reconstruct a species tree of the genus Heliconius using the full-likelihood multispecies coalescent, an exciting approach for genera with a history of extensive gene flow and introgression. With this, they obtain a species tree with H. aoede as the earliest diverging lineage, in sync with ecological and morphological characters. They also add resolution to the species relationships of the melpomene-silvaniform clade and quantify introgression events. Finally, they trace the origins of an inversion on chromosome 15 that exists as a polymorphism in H. numata, but is fixed in other species. Overall, obtaining better species tree resolutions and estimates of gene flow in groups with extensive histories of hybridization and introgression is an exciting avenue. Being able to control for ILS and get estimates between sister species are excellent perks. One overall quibble is that the paper seems to be best suited to a Heliconius audience, where past trees are easily recalled, or members of the different clades are well known.

Overall, applying approaches such as these to gain greater insight into species relationships with extensive gene flow could be of interest to many researchers.

---

## [Referee Report · Reviewer #3 (Public Review)]

The authors use a full-likelihood multispecies coalescent (MSC) approach to identify major introgression events throughout the radiation of Heliconius butterflies, thereby improving estimates of the phylogeny. First, the authors conclude that H. aoede is the likely outgroup relative to other Heliconius species; miocene introgression into the ancestor of H. aoede makes it appear to branch later. Topologies at most loci were not concordant with this scenario, though 'aoede-early' topologies were enriched in regions of the genome where interspecific introgression is expected to be reduced: the Z chromosome and larger autosomes. The revised phylogeny is interesting because it would mean that no extant Heliconius species has reverted to a non-pollen-feeding ancestral state. Second, the authors focus on a particularly challenging clade in which ancient and ongoing gene flow is extensive, concluding that silvaniform species are not monophyletic. Building on these results, a third set of analyses investigates the origin of the P1 inversion, which harbours multiple wing patterning loci, and which is maintained as a balanced polymorphism in H. numata. The authors present data supporting a new scenario in which P1 arises in H. numata or its ancestor and is introduced to the ancestor of H. pardilinus and H. elevatus - introgression in the opposite direction to what has previously been proposed using a smaller set of taxa and different methods.

The analyses were extensive and methodologically sound. Care was taken to control for potential sources of error arising from incorrect genotype calls and the choice of a reference genome. The argument for H. aoede as the earliest-diverging Heliconius lineage was compelling, and analyses of the melpomene-silvaniform clade were thorough.

The authors have demonstrated the strengths of a full-likelihood MSC approach when reconstructing the evolutionary history of "difficult" clade. This approach, however, can quickly become intractable in large species complexes where there is extensive gene flow or significant shifts in population size. In such cases, there may be hundreds of potentially important parameters to estimate, and alternate introgression scenarios may be impossible to disentangle. This is particularly challenging in systems unlike Heliconius, where fewer data are available and there is little a priori knowledge of reproductive isolation, genome evolution, and the unique life history traits of each species.

---

## [Author Response]

The following is the authors’ response to the original reviews.

**Public Reviews:**

**Reviewer #1 (Public Review):**
Summary:This study aims to further resolve the history of speciation and introgression in Heliconius butterflies.The authors break the data into various partitions and test evolutionary hypotheses using the Bayesian software BPP, which is based on the multispecies coalescent model with introgression. By synthesizing these various analyses, the study pieces together an updated history of Heliconius, including a multitude of introgression events and the sharing of chromosomal inversions.Strengths:Full-likelihood methods for estimating introgression can be very computationally expensive, making them challenging to apply to datasets containing many species. This study provides a great example of how to apply these approaches by breaking the data down into a series of smaller inference problems and then piecing the results together. On the empirical side, it further resolves the history of a genus with a famously complex history of speciation and introgression, continuing its role as a great model system for studying the evolutionary consequences of introgression. This is highlighted by a nice Discussion section on the implications of the paper's findings for the evolution of pollen feeding.Weaknesses:The analyses in this study make use of a single method, BPP. The analyses are quite thorough so this is okay in my view from a methodological standpoint, but given this singularity, more attention should be paid to the weaknesses of this particular approach.

In the Discussion, we have now added a discussion of the limitations of our approach in the section 'Approaches for estimating species phylogeny with introgression from whole-genome sequence data: advantages and limitations.'

Additionally, little attention is paid to comparable methods such as PhyloNet and their strengths and weaknesses in the Introduction or Discussion.

We have also mentioned other methods (PhyloNet and starBEAST) in our Discussion. Our attempts to obtain usable estimates from PhyloNet were unsuccessful. In another study, the full likelihood version of PhyloNet (comparable in intent to the BPP methodology used here) could run with only small datasets of ~100 loci; see Edelman et al. (2019).

BPP reduces computational burden by fixing certain aspects of the parameter space, such as the species tree topology or set of proposed introgression events. While this approach is statistically powerful, it requires users to make informed choices about which models to test, and these choices can have downstream consequences for subsequent analyses. It also might not be as applicable to systems outside of Heliconius where less previous information is available about the history of speciation and introgression. In general, it is likely that most modelling decisions made in the study are justified, but more attention should be paid to how these decisions are made and what the consequences of them could be, including alternative models.

We agree with the reviewer that inferring the species tree topology and placing introgression events on the species tree, although well justified here, may be challenging in many groups of organisms and may affect downstream analyses. We now discuss this as a limitation of our approach in the Discussion. In general, the initial MSC analysis without gene flow should provide information about possible species trees and introgression events. We can construct multiple introgression models and perform parameter estimation and model comparison to decide which best fits the data. This is summarized in the last paragraph of the section 'Approaches for estimating species phylogeny with introgression from whole-genome sequence data: advantages and limitations.' It would, of course, be nice to have a completely unsupervised method that could work with large phylogenies, but this is currently computationally impossible.

• Co-estimating histories of speciation and introgression remains computationally challenging. To circumvent this in the study, the authors first estimate the history of speciation assuming no gene flow in BPP. While this approach should be robust to incomplete lineage sorting and gene tree estimation, it is still vulnerable to gene flow. This could result in a circular problem where gene flow causes the wrong species tree to be estimated, causing the true species tree to be estimated as a gene flow event.

The goal of this initial analysis is to obtain a list of possible species trees with introgression events. We assume that gene flow results in a topology that is informative about the lineages involved. We also focus on common MAP trees with high posterior probabilities as less frequent trees or trees with low posterior probabilities reflect high uncertainty and are more likely to be erroneous. A difficulty is to decide which tree topology is most likely to be the true species tree. We summarize our approach in the Discussion.

This is a flaw that this approach shares with summary-statistic approaches like the D-statistic, which also require an a-priori species tree.

In a sense, this is true, but BPP is more flexible because it can be used to explore an arbitrary introgression model on any type of tree, while summary methods like D-statistic assume a specific species phylogeny with a particular introgression between nonsister lineages as well as fixed sampling configurations. Furthermore, as shown in the paper, we can compare different assumed trees, and test between them; we do this repeatedly in the paper for difficult branch placement issues. In contrast, summary methods such as the D-statistic works with species quartets only and do not work with either smaller or larger species trees.

Enrichment of particular topologies on the Z chromosome helps resolve the true history in this particular case, but not all datasets will have sex chromosomes or chromosome-level assemblies to test against.

Yes, we have the privilege of having chromosome-level assemblies available for Heliconius. In general, a spatial pattern of species tree estimates across genomic blocks can be informative about possible topologies that could represent the true species relationship. Then these candidate species trees can be tested by fitting different introgression models (as in Figure 1D,E) or by using the recombination rate argument (Figure 1F), which prefers trees common in low recombination rate regions of the genome, although this requires knowing a recombination rate map. In our case, we used a chromosome-level recombination rate per base pair, which is negatively correlated with the chromosome size. We have clarified this in the text. Ultimately, multiple lines of evidence should be examined before deciding on the most likely species tree. We now mention these potential difficulties with applying our methods to other datasets as limitations of our approach in the Discussion.

• The a-priori specification of network models necessarily means that potentially better-fitting models to the data don't get explored. Models containing introgression events are proposed here based on parsimony to explain patterns in gene tree frequencies. This is a reasonable and common assumption, but parsimony is not always the best explanation for a dataset, as we often see with phylogenetic inference. In general, there are no rigorous approaches to estimating the best-fitting number of introgression events in a dataset.

Joint inference of species topologies and possible introgression events remains computationally challenging. PhyloNet implements this joint inference but is limited to small datasets (<100 loci) and we found it to be unreliable.

Likewise, the study estimates both pulse and continuous introgression models for certain partitions, though there is no rigorous way to assess which of these describes the data better.

The Bayes factor can be used to compare different models fitted to the same data, for example, different MSC-I models with different introgression events, or MSC-I models with gene flow in pulses versus MSC-M models with continuous gene flow. We did not attempt this as it was clear to us that a better model would include both modes of gene flow, but such an option is not currently implemented in any software. Rather, we relied on our exploratory analysis (BPP MSC and 3s) and previous knowledge to inform a likely introgression model. In the case of groups that we fitted the MSC-M models, we chose to provide an intuitive justification as to why they might be more realistic than the MSC-I model without formally performing model selection.

• Some aspects of the analyses involving inversions warrant additional consideration. Fewer loci were able to be identified in inverted regions, and such regions also often have reduced rates of recombination. I wonder if this might make inferences of the history of inverted regions vulnerable to the effects of incomplete lineage sorting, even when fitting the MSC model, due to a small # of truly genealogically independent loci.

We agree with the reviewer that it is challenging to infer the history of a small region of the genome, such as the inversions studied here. Indeed, the presence of only a few loci in the 15b inversion means there is only limited information in the data for the species tree, as reflected in the low posterior probabilities for the MAP tree (Figure 3A). The effect of using tightly linked loci in the inversion should be increased uncertainty in the estimates, but not a systematic bias towards any particular species tree topology. Since major patterns of species relationships in each of the 15a, 15b and 15c regions are clear, we do not expect these effects to strongly influence our conclusions.

Additionally, there are several models where introgression events are proposed to explain the loss of segregating inversions in certain species. It is not clear why these scenarios should be proposed over those in which the inversion is lost simply due to drift or selection.

We know that the 15b inversion is absent in most species except for H. numata and H. pardalinus, at least, and that introgression of the inversion occurred between these two species, based on previous studies such as Jay et al (2018) and our own analysis. Polymorphism at this inversion forms a well-known “supergene” that affects mimicry, and is maintained by documented balancing selection in H. numata. Given this information, we propose a few possible scenarios of how the inversion might have originated, and when and where the introgression might have occurred, shown in Figure 3. In particular, the direction of introgression is something we test specifically. One way to test among these scenarios is to date the origin and introgression event of the inversion, but doing so properly is beyond the scope of this work. Nonetheless, we argue that it is at least likely that one difference between H. pardalinus and its sister species H. elevatus is the presence of the 15b inversion. Since other evidence shows that colour patterning loci in H. elevatus originated from an unrelated species, H. melpomene (i.e. the 15b and other non-inverted colour patterning loci), it is indeed likely that the inversion was “swapped out” by an uninverted sequence from H. melpomene during the formation of H. elevatus.

We are aware that hypotheses such as these might appear highly elaborate and unparsimonious. But these are the conclusions where the data lead us. In the melpomene-silvanform clade, many speciation and introgression events occurred in short succession, and wild-caught hybrids prove that occasional hybridizations can occur across all 15 or so species in the group. We now detail how we have looked only for the major introgression patterns using a limited number of key speces. We leave fuller analyses for future work.

In the main text, we have revised our discussion of the four proposed scenarios for 15b to improve clarity. We have also updated the introgression model from the melpomene-cydno clade to H. elevatus to be unidirectional based on the BPP results in Figure S18.

**Reviewer #2 (Public Review):**
Thawornwattana et al. reconstruct a species tree of the genus Heliconius using the full-likelihood multispecies coalescent, an exciting approach for genera with a history of extensive gene flow and introgression. With this, they obtain a species tree with H. aoede as the earliest diverging lineage, in sync with ecological and morphological characters. They also add resolution to the species relationships of the melpomene-silvaniform clade and quantify introgression events. Finally, they trace the origins of an inversion on chromosome 15 that exists as a polymorphism in H. numata, but is fixed in other species. Overall, obtaining better species tree resolutions and estimates of gene flow in groups with extensive histories of hybridization and introgression is an exciting avenue. Being able to control for ILS and get estimates between sister species are excellent perks. One overall quibble is that the paper seems to be best suited to a Heliconius audience, where past trees are easily recalled, or members of the different clades are well known.

We thank the reviewer for the accurate summary and positive comments. Although our data and some of the discussion are specific to Heliconius, we believe our analysis framework will be useful to study species phylogeny and introgression in other taxa as well.

Overall, applying approaches such as these to gain greater insight into species relationships with extensive gene flow could be of interest to many researchers. However, the conclusions could be strengthened with a bit more clarity on a few points.1. The biggest point of concern was the choice of species to use for each analysis. In particular the omission of H. ismenius in the resolution of the BNM clade species tree. The analysis of the chromosome 15 inversion seems to rely on the knowledge that H. ismenius is sister to H. numata, so without that demonstrated in the BNM section the resulting conclusions of the origin of that inversion are less interruptible.

The choice of species to be included was mainly based on available high-quality genome resequence data from Edelman et al (2019), which were chosen to cover most of the major lineages within the genus. We agree that inclusion of H. ismenius would strengthen the analysis of the melpomene-silvaniform clade. In particular, it would be interesting to know which of only H. numata or H. numata+H. ismenius are responsible for the main source of genealogical variation across the genome in this group in Figure 2. The reviewer is correct in saying that we do assume that H. ismenius and H. numata are sister species. This relationship is supported by our analysis (Figure 3A) and previous analyses of genomic data, e.g. Zhang et al (2016), Cicconardi et al. (2023) and Rougemont et al. (2023). We made this clearer in the text:

"Although this conclusion assumes that H. numata and H. ismenius are sister species while H. ismenius was not included in our species tree analysis of the melpomene-silvaniform clade (Figure 2), this sister relationship agrees with previous genomic studies of the autosomes and the sex chromosome (Zhang et al. 2016; Cicconardi et al. 2023; Rougemont et al. 2023)."

1. An argument they make in support of the branching scenario where H. aoede is the earliest diverging branch is based on which chromosomes support that scenario and the key observation that less introgression is detected in regions of low recombination. Yet, they go no further to understand the relationship between recombination rate and species trees produced.

We believe Figure 1F does examine this relationship, showing that trees under scenario 2 are more common in regions of the genome with lower recombination rates (i.e. in longer chromosomes). We added more clarification in the text where Figure 1F is mentioned. The relationship between recombination and introgression in Heliconius was earlier discovered and shown using windowed estimated gene trees in Martin et al. (2019) and in Edelman et al. (2019), so we did not re-test this here.

1. How the loci were defined could use more clarity. From the methods, it seems like each loci could vary quite a bit in total bp length and number of informative sites. Understanding the data processing would make this paper a better resource for others looking to apply similar approaches.

We added a new supplemental figure, Figure S20, to illustrate how coding and noncoding loci were extracted from the genome.

**Reviewer #3 (Public Review):**
The authors use a full-likelihood multispecies coalescent (MSC) approach to identify major introgression events throughout the radiation of Heliconius butterflies, thereby improving estimates of the phylogeny. First, the authors conclude that H. aoede is the likely outgroup relative to other Heliconius species; miocene introgression into the ancestor of H. aoede makes it appear to branch later. Topologies at most loci were not concordant with this scenario, though 'aoede-early' topologies were enriched in regions of the genome where interspecific introgression is expected to be reduced: the Z chromosome and larger autosomes. The revised phylogeny is interesting because it would mean that no extant Heliconius species has reverted to a non-pollen-feeding ancestral state. Second, the authors focus on a particularly challenging clade in which ancient and ongoing gene flow is extensive, concluding that silvaniform species are not monophyletic. Building on these results, a third set of analyses investigates the origin of the P1 inversion, which harbours multiple wing patterning loci, and which is maintained as a balanced polymorphism in H. numata. The authors present data supporting a new scenario in which P1 arises in H. numata or its ancestor and is introduced to the ancestor of H. pardilinus and H. elevatus - introgression in the opposite direction to what has previously been proposed using a smaller set of taxa and different methods.The analyses were extensive and methodologically sound. Care was taken to control for potential sources of error arising from incorrect genotype calls and the choice of a reference genome. The argument for H. aoede as the earliest-diverging Heliconius lineage was compelling, and analyses of the melpomene-silvaniform clade were thorough.The discussion is quite short in its current form. In my view, this is a missed opportunity to summarise the level of support and biological significance of key results. This applies to the revised Melpomenesilvaniform phylogeny and, in particular, the proposed H. numata origin of P1. It would be useful to have a brief overview of the relationships that remain unclear, and which data (if any) might improve estimates.

We added a paragraph in the Discussion to summarize our key findings in 'An updated phylogeny of Heliconius', and discuss issues that remain uncertain.

It was good to see the authors reflect on the utility of full-likelihood approaches more generally, though the discussion of their feasibility and superiority was at times somewhat overstated and reductive. Alternative MSC-based methods that use gene tree frequencies or coalescence times can be used to infer the direction and extent of introgression with accuracy that is satisfactory for a wide variety of research questions. In practice, a combination of both approaches has often been successful. Although full-likelihood approaches can certainly provide richer information if specific parameter estimates are of interest, they quickly become intractable in large species complexes where there is extensive gene flow or significant shifts in population size. In such cases, there may be hundreds of potentially important parameters to estimate, and alternate introgression scenarios may be impossible to disentangle. This is particularly challenging in systems, unlike Heliconius where there is little a priori knowledge of reproductive isolation, genome evolution, and the unique life history traits of each species. It would be useful for the authors to expand on their discussion of strategies that can simplify inference problems in such systems, acknowledging the difficulties therein.

We agree that approximate methods based on summary statistics (e.g. gene tree topologies) are computationally much cheaper and are sometimes useful. We now discuss limitations of our approach regarding strategies for constructing possible introgression models, computational cost and analysis of large phylogenies, and modeling assumptions in the MSC framework in the first section of the Discussion.

**Reviewer #1 (Recommendations For The Authors):**
In addition to the comments raised in the public review, I have some minor suggestions:In the Introduction, "Those methods have limited statistical power" implies summary-statistic methods have a high false negative rate for inferring the presence of introgression, which I don't think is true.

We removed 'statistical' as we used the term power loosely to mean ability to estimate more parameters in the model by making a better use of information in the sequence data and not in the sense of a true positive rate.

When discussing full-likelihood approaches in a general sense, please cite additional methods than just BPP, such as PhyloNet.

We added references for PhyloNet (Wen & Nakhleh, 2018) and starBEAST (Zhang et al., 2018) in the Introduction and Discussion.

Consider explicitly labelling chromosomal region 21 as the Z chromosome in relevant Figures, for ease of interpretation.

In the main figures, we changed the chromosome label from 21 to Z.

From reading the main text it's not clear what a "3s analysis" is

The 3s analysis estimates pairwise migration rates between two species by fitting an MSC-withmigration (MSC-M) model, also known as isolation-with-migration (IM), for three species, where gene flow is allowed between the two sister species while the outgroup is used to improve the power but does not involved in gene flow. We changed the text from

"We use estimates of migration rates between each pair of species with a 3s analysis under the IM model of species triplets ..."

to

"We use estimates of migration rates between each pair of species under the the MSC-withmigration (MSC-M or IM) model of species triplets (3s analysis) ..."

"This agrees with the scenario in which H. elevatus is a result of hybrid speciation between H.pardalinus and the common ancestor of the cydno-melpomene clade [42, 43]." I don't think this model provides any support for hybrid speciation in particular, over a standard post-speciation introgression scenario.

We took the finding that the introgression from the melpomene-cydno clade into H. elevatus occurs almost right after H. elevatus split off from H. pardalinus as evidence for hybrid speciation. We revised the text to make this clearer:

"Our finding that divergence of H. elevatus and introgression from the cydno-melpomene clade occurred almost simultaneously provides evidence for a hybrid speciation origin of H. elevatus resulting from introgression between H. pardalinus and the common ancestor of the cydno-melpomene clade (Rosser et al. 2019; Rosser et al. 2023)."

In particular, the Rosser et al. (2023) paper has now been submitted, and is the main paper to cite for the hybrid speciation hypothesis for H. elevatus.

"while clustering with H. elevatus would suggest the opposite direction of introgression" careful with terminology here; is this about direction (donor vs. recipient species) or taxa involved (which is not direction)?

This is about the direction of introgression, not the taxa involved. We modified the text to make this clearer:

"By including H. ismenius and H. elevatus, sister species of H. numata and H. pardalinus respectively, different directions of introgression should lead to different gene tree topologies. Clustering of (H. numata with the inversion, H. pardalinus) with H. numata without the inversion would suggest the introgression is H. numata → H. pardalinus while clustering of (H. numata with the inversion, H. pardalinus) with H. elevatus would suggest H. pardalinus → H. numata introgression."

**Reviewer #3 (Recommendations For The Authors):**
The work is methodologically sound and rigorous but could have been reported and discussed with greater clarity.It was difficult to assess the level of support for the proposed P1 introgression scenario without digging through the extensive supplementary materials. The discussion would ideally be used to clarify and summarise this.

We have substantially revised the section on the P1 inversion. We also mention in the Results (in the final paragraph of the inversion section) and Discussion that our data provided robust evidence that the introgression of the inversion is from H. numata into H. pardalinus while its precise origin (in which lineage and when it originated) remains uncertain.

The authors may also wish to compare their results to the recent work by Rougemont et al. on introgression between H. hecale and H. ismenius in the discussion.

We now mention Rougemont et al. (2023) in the Discussion as an example of introgression of small regions of the genome involved in wing patterning. We also acknowledge that our updated phylogeny does not include this kind of local introgression.

It was not initially obvious which number corresponded to the Z chromosome in any of the figures, even though this is critical to their interpretation.

We changed the label for chromosome 21 to Z in the main figures.

The supplementary tables should be described in more detail. For example, what is 'log_bf_check' and 'prefer_pred' in supplementary table S11?

We added more details explaning necessary quantities in the table heading in both SI file and in the spreadsheet.

Minor comments:First paragraph of 'Complex introgression in the 15b inversion region (P locus):' Rephrase "This suggests another introgression between the common...".

We modified the text as follows:

"Another feature of this 15b region is that among the species without the inversion, the cydnomelpomene clade clusters with H. elevatus and is nested within the pardalinus-hecale clade (without H. pardalinus). This is contrary to the expectation based on the topologies in the rest of the genome (Figure 2A, scenarios a–c) that the cydno-melpomene clade would be sister to the pardalinus-hecale clade (without H. pardalinus). One explanation for this pattern is that introgression occurred between the common ancestor of the cydno-melpomene clade and either H. elevatus or the common ancestor of H. elevatus and H. pardalinus together with a total replacement of the non-inverted 15b in H. pardalinus by the P1 inversion from H. numata (Jay et al. 2018). We confirm and quantify this introgression below."

Second paragraph of 'Major Introgression Patterns in the melpomene-silvaniform clade:' "cconclusion" should be "conclusion."

Corrected.

Paragraph preceding discussion: sentences toward the end of the paragraph should be rephrased for clarity. E.g. "different tree topologies are expected under different direction of introgression."

We revised this paragraph. The sentence now says:

"By including H. ismenius and H. elevatus, sister species of H. numata and H. pardalinus respectively, different directions of introgression should lead to different gene tree topologies.

Clustering of (H. numata with the inversion, H. pardalinus) with H. numata without the inversion would suggest the introgression is H. numata → H. pardalinus while clustering of (H. numata with the inversion, H. pardalinus) with H. elevatus would suggest H. pardalinus → H. numata introgression."

I enjoyed reading this paper and I am certain it will generate discussion and future research.